# GMM-TS: Gating Architecture for Multi-Modal Time Series Forecasting

## Abstract

Forecasting future trends in complex domains often requires leveraging diverse data sources beyond traditional numerical time series. However, integrating heterogeneous data types into a unified forecasting framework remains an underexplored challenge. Existing multi-modal time series forecasting approaches often employ static and simplistic fusion mechanisms or yield non-interpretable representations with a limited modularity. We propose GMM-TS, a learnable gating architecture, inspired by mixture-of-experts, which dynamically integrates predictions from multiple uni-modal experts, each specialized in a distinct modality (e.g., text or numerical signals). Our method computes per-time-step expert weights using a Transformer Encoder. This enables fine-grained, interpretable fusion of multiple experts (two or more) and supports both joint and offline training modes. Extensive evaluations show that GMM-TS consistently outperforms state-of-the-art baselines across nine domains, multiple forecast horizons, and various expert configurations. We also include, for the first time, to the best of our knowledge, the option to integrate more than two experts. Our framework is efficient, extensible, and inherently interpretable. Code will be released upon acceptance.

## 1 Introduction

Forecasting future trends from time series data is a fundamental challenge across domains such as finance, healthcare, climate modeling, and transportation (e.g., Choi et al. (2022); Castán-Lascorz et al. (2022)). Traditional models—ranging from ARIMA and exponential smoothing to deep learning methods—primarily operate on numerical inputs and assume that past observations are sufficient to predict the future (Lim & Zohren, 2021). Yet, in many real-world scenarios, critical contextual information resides in other modalities: textual reports (Liu et al., 2024a), event logs (Hong et al., 2024), or visual summaries (Daswani et al., 2024) often contain signals that precede or explain changes in time series trends. This motivates the development of forecasting systems that can effectively leverage multi-modal inputs.

Recent work has begun to explore multi-modal time series forecasting (TSF). TimeMMD (Liu et al., 2024a) introduced a multi-domain benchmark and a plug-and-play architecture for combining numerical and textual signals which relies on a fixed, manually tuned weight during inference. GPT4MTS (Jia et al., 2024) takes a different approach, proposing a soft-prompting strategy that jointly encodes numerical and textual input within a GPT-2 decoder. While these approaches highlight the potential of multi-modal inputs, they suffer from key limitations. First, their fusion mechanisms are either fixed (e.g., static weights) or implicit (e.g., prompting within LLMs), offering no clear way to adapt to the input or interpret how modalities contribute to the prediction. Second, they lack modularity: they do not support flexible configuration of forecasting models, making it difficult to swap, mix, or scale expert architectures. Finally, they are restricted to binary fusion, preventing the integration of more than two experts or modalities. These shortcomings limit their practical utility in diverse, evolving real-world forecasting settings.

Recent surveys (Liang et al., 2024) identify a core limitation of current multi-modal TSF methods: the absence of a clear, learnable mechanism for effectively fusing heterogeneous data sources in an interpretable and adaptive manner. We address this challenge with **GMM-TS**, a modular gating architecture that enables dynamic and interpretable fusion of multiple uni-modal TSF models ("experts"). At each forecast step, the model predicts expert-specific weights conditioned on a com-

bination of expert latents, a learnable gating token, and the raw input. This design allows GMM-TS to adaptively prioritize the most relevant experts over time while enhancing their joint performance.

GMM-TS is uniquely flexible, supporting fusion over arbitrary sets of experts (e.g., triplets, quartets) spanning different modalities, and accommodates both end-to-end joint training and offline expert pre-training for efficient reuse across tasks and domains. Our approach is motivated by Mixture-of-Experts (MoE) principles (Jacobs et al., 1991), where uni-modal inputs are used to assign single weights to architecturally identical experts. However, unlike conventional MoE architectures, which are not directly suited for multi-modal TSF, GMM-TS introduces a novel gating mechanism tailored to heterogeneous inputs and expert types, learning from both raw inputs and intermediate expert representations.

We evaluate GMM-TS on the TimeMMD benchmark across nine domains, four forecast horizons, and multiple expert combinations. Our method outperforms strong baselines, including TimeMMD and GPT4MTS, across all tested settings. Experiments demonstrate the benefit of our fusion strategy, including superior performance when aggregating more than two experts. Ablation studies validate the robustness of the gating mechanism, comparing alternative aggregation strategies (e.g., latent and hierarchical fusion) and varying the gating dimension. Finally, the gating network produces explicit per-expert weight distributions per time step, offering fine-grained interpretability for downstream analysis or decision support.

In summary, our main contributions are:

- We propose **GMM-TS**, a **Transformer-based gating architecture for multi-modal time series forecasting** that learns to dynamically weigh predictions from any number of uni-modal experts.

- We show that GMM-TS **outperforms state-of-the-art multi-modal and uni-modal baselines** across domains, horizons, and expert configurations, including settings with three or more experts.

- We demonstrate the **interpretability and flexibility** of our approach, offering explicit per-expert attribution and support for both joint and offline training strategies.

## 2 RELATED WORK

**Uni-modal time series forecasting.** Time series forecasting (TSF) has been extensively studied in the uni-modal setting, where models rely solely on numerical inputs. Classical statistical methods such as ARIMA (Nelson, 1998) and exponential smoothing have been widely used, but recent progress has been driven by deep learning architectures such as MLPs (Yi et al., 2024; Chen et al., 2023), temporal CNNs (Wu et al., 2023), and Transformer-based models (Nie et al., 2022b; Kitaev et al., 2020; Zhou et al., 2021; Wu et al., 2021b; Zhou et al., 2022c). These methods assume that future behavior can be inferred solely from past numerical values, which limits their effectiveness in complex, event-driven domains.

**Multi-modal time series forecasting.** To address these limitations, recent work has explored incorporating additional modalities. Among peer-reviewed efforts, TimeMMD (Liu et al., 2024a) and GPT4MTS (Jia et al., 2024) remain the only systems, to the best of our knowledge, that combine numerical and textual signals for time series forecasting. Notably Liu et al. (2024b) highlighted exogenous multi-modal TSF as an *emerging research direction*, with limited existing work - a gap our paper directly addresses. TimeMMD provides a benchmark and a plug-and-play architecture with a modular expert selection. However, its fusion mechanism is static (fixed weight) and not input-adaptive and it is limited to using only two experts: one numerical and one textual. GPT4MTS adapts a GPT2 decoder with soft prompts to jointly encode text and numeric features, offering an end-to-end architecture but lacking modularity and interpretability.

We distinguish the challenge of fusing *exogenous modalities*, such as numerical time series and text descriptions of events, from the fusion of a time series with representations derived from it. The former is the focus of our work and of works like GPT4MTS and TimeMMD, while the latter represents a different class of fusion problems. An example for non-exogenous multi-modal TSF is Time-VLM (Zhong et al., 2025), a recently introduced tri-modal fusion approach using vision-language models to process time series plots derived from numerical time series. We note that Time-VLM employs shallow concatenation for fusion and does not support dynamic, per-time-step

weighting, or modular expert setup (replacing and adding experts), in contrast to our proposed gating mechanism. For a detailed comparison with prior methods, see Table 6 in Appendix A.

**Mixture-of-Experts and gating.** Our method builds on Mixture-of-Experts (MoE) concepts (Jacobs et al., 1991), where a gating network assigns input-specific weights to expert predictions. While early MoE models used homogeneous expert sets, more recent variants such as MERA (Zhou et al., 2025) use retrieval-augmented gating to model diversified behaviors in numerical stock prediction. However, MERA is limited to uni-modal inputs and lacks per-expert interpretability. Our proposed architecture generalizes these ideas to the multi-modal setting by supporting heterogeneous experts, combining raw input tokens and expert latents via Transformer-based attention, and producing explicit, interpretable weights for each expert and forecast step.

## 3 METHOD

We present **GMM-TS**, a Transformer-based gating architecture for multi-modal time series forecasting. Here, "gating" refers to the adaptive weighting mechanism inspired by Mixture-of-Experts (MoE) models Jacobs et al. (1991); Shazeer & et al. (2017). GMM-TS combines predictions from individual expert models, where each expert specializes in processing either numerical time series data (TSF-N) or textual data (TSF-T). As shown in Fig. 1, our model learns to assign per-time-step weights to each expert's prediction based on both the raw numerical input (i.e., values observed before the forecast horizon) and the latent representations learned by the experts. The uni-modal experts and the gating network are jointly trained in an end-to-end fashion.

This joint optimization enables the experts to specialize in ways that are informed by the gating dynamics, in contrast to prior multi-modal TSF methods which rely on staged training or static fusion rules. We further explore alternative fusion strategies and training configurations in Section 4.3, and analyze their impact through ablation studies in Section 4.4.

### 3.1 UNI-MODAL TSF EXPERTS

We formulate the time series forecasting (TSF) task as predicting a sequence of future values $Y \in \mathbb{R}^{p \times d_V}$ after a reference time point $t^*$, based on observations collected prior to $t^*$. Here, $p$ denotes the forecast horizon (i.e., number of future time steps), and $d_V$ is the dimensionality of the target variable at each step. Following the widely adopted channel independence assumption Nie et al. (2022b), we set $d_V = 1$ and treat each target variable independently, such that $Y \in \mathbb{R}^p$.

Input observations may come from a variety of modalities, including numerical, textual, or visual signals. In this work, we focus on numerical and textual inputs, but our framework generalizes to other modalities. We refer to experts trained on numerical time series as *TSF-N experts*, and those trained on textual time series as *TSF-T experts*.

**TSF-N experts.** A TSF-N expert processes a multivariate numerical time series $X_n \in \mathbb{R}^{l_n \times d_n}$, where $l_n$ is the lookback window (number of past time steps), and $d_n$ is the number of variables. A typical TSF-N model operates in two stages (illustrated in Fig. 1, top left):

$$h_n = B_n(X_n), \qquad Y = f_n(h_n) \tag{1}$$

where $B_n$ is a neural backbone that encodes the input into a latent representation $h_n$, and $f_n$ is a fully connected head that outputs the forecast. Common choices for $B_n$ include temporal convolutional networks, MLPs, and attention-based models such as Transformers Wang et al. (2024); Nie et al. (2022b); Wu et al. (2021b). TSF-N experts are trained end-to-end using supervised loss on the prediction error.

**TSF-T experts.** Text-based forecasting models, or TSF-T experts, operate on sequences of textual inputs that describe temporal dynamics. Let $X_t = [x_t^1, \ldots, x_t^{l_t}]$ denote a sequence of text observations collected prior to forecast time $t^*$, where $l_t$ is the textual lookback window. Following the formulation introduced in the TimeMMD framework Liu et al. (2024a), we use a frozen large language model (LLM) $LM$ that is prompted to summarize the textual history and produce a forecast embedding. Specifically, the LLM processes the prompt to generate a representation $h_t^{LM}$, which

is then projected into a task-specific latent space by an MLP $B_t$, and decoded by a fully connected layer $f_t$:

$$h_t^{LM} = LM(X_t), \qquad h_t = B_t(h_t^{LM}), \qquad Y = f_t(h_t) \tag{2}$$

Only $B_t$ and $f_t$ are trained; the LLM $LM$ remains frozen. This design enables efficient adaptation to forecasting tasks through lightweight training, while leveraging rich pretrained representations via prompting. The TSF-T expert architecture is illustrated in Fig. 1 (bottom left).

## 3.2 Multi-modal TSF via gated fusion of uni-modal expert predictions

### 3.2.1 Model architecture

Our architecture is composed of uni-modal experts and a Transformer-based gating network, as illustrated in Fig. 1. Let $E_n = \{e_n^i\}_{i=1}^{k_n}$ and $E_t = \{e_t^i\}_{i=1}^{k_t}$ denote sets of TSF-N and TSF-T experts, respectively (shown in blue and orange in Fig. 1). Let $E = E_n \cup E_t$ be the complete set of uni-modal experts. Our goal is to learn a dynamic, input-dependent fusion strategy that combines expert predictions to maximize forecasting accuracy.

Given a pair of inputs $(X_n, X_t)$ a numerical and textual time series—we extract latent representations from each expert in $E_n$ and $E_t$, yielding two sets of embeddings: $H_n = \{h_n^i\}_{i=1}^{k_n}$ and $H_t = \{h_t^i\}_{i=1}^{k_t}$. Since the latent representations from each expert may differ in dimension, we project them into a shared latent space of dimension $d_m$ using expert-specific fully connected layers $f_e$, for each $e \in E$. These projected embeddings, shown in dark green in Fig. 1, form the unified set:

$$H_m = \{h_m^i\}_{i=1}^{|E|}, \quad h_m^i \in \mathbb{R}^{d_m}$$

We concatenate the projected vectors into a matrix $H_s \in \mathbb{R}^{d_m \times |E|}$, where each column corresponds to one expert. To this matrix, we prepend two additional inputs:

- An *input token* $x \in \mathbb{R}^{d_m}$, representing the raw numerical input $X_n$, processed via patching, temporal and positional encoding, average pooling, and projection (visualized in pink in Fig. 1);
- A learnable *gating token* $g \in \mathbb{R}^{d_m}$, which controls the expert weighting process (shown in light green in Fig. 1).

The full sequence fed to the gating module is $S = [g, x, H_s]^\top \in \mathbb{R}^{(|E|+2) \times d_m}$, where the transpose ensures token-major format. We apply a Transformer Encoder to $S$, which uses multi-head self-attention to model interactions between experts, input signals, and the gating context. Let $g_o \in \mathbb{R}^{d_m}$ denote the output at the position corresponding to the gating token.

We pass $g_o$ through an MLP head followed by a Softmax to obtain a weight matrix $W \in \mathbb{R}^{p \times |E|}$, where each row $W[t]$ represents the expert weight distribution at forecast step $t$. Let $Y_E \in \mathbb{R}^{p \times |E|}$ denote the predictions from all experts. The final forecast is computed as a weighted sum across experts at each time step:

$$Y[t] = \sum_{j=0}^{|E|-1} W[t, j] \cdot Y_E[t, j], \quad \text{for } t = 0, \dots, p-1 \tag{3}$$

with $Y_E \in \mathbb{R}^{p \times |E|}$, the predictions made by the uni-modal experts in $E$.

### 3.2.2 Training and inference

We jointly train the uni-modal experts and the gating network in an end-to-end manner using the mean squared error (MSE) loss:

$$\text{MSE} = \frac{1}{N} \sum_{i=1}^{N} (y_i - \hat{y}_i)^2 \tag{4}$$

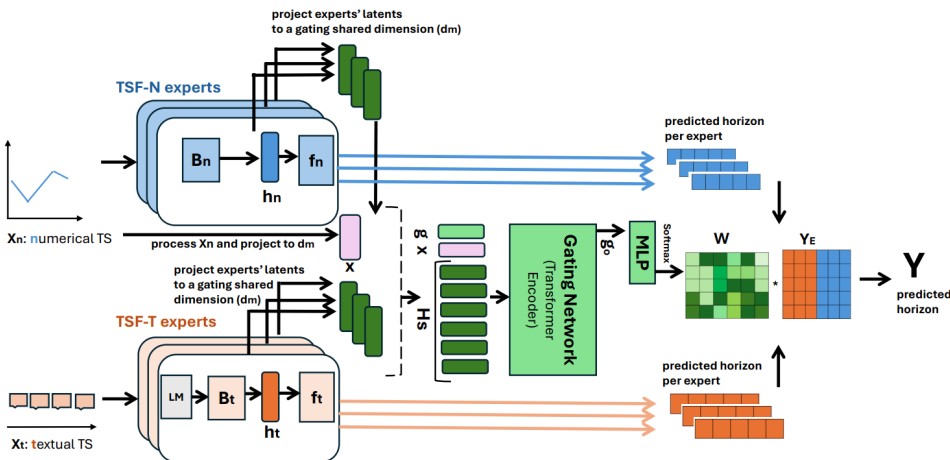

Figure 1: Overview of the GMM-TS architecture. TSF-N (numerical) and TSF-T (textual) experts (blue and orange, respectively) produce latent representations, which are projected into a shared latent space (dark green). A Transformer-based gating network fuses these latents, along with an input token (pink) summarizing the raw numerical input (time steps prior to the forecast horizon) and a learnable gating token (light green), to produce dynamic expert weights per forecast time step. The final prediction is a weighted sum over all expert outputs.

where $N$ is the number of training examples, $y_i$ is the ground-truth target, and $\hat{y}_i$ is the predicted value for the $i$-th instance. For TSF-T experts, we freeze the pretrained LLM backbone and train only the downstream MLP projection and prediction layers.

During inference, given a new input pair $(X_n, X_t)$, we compute both latent representations and forecast outputs from all experts. These are passed to the gating network, which produces expert weights and aggregates the outputs into a final fused forecast as described in Section 3.2.1.

## 4 EXPERIMENTS

In this section, we empirically evaluate the effectiveness of **GMM-TS**. Our experimental protocol and implementation details are described in Section 4.1.

**Key findings:** our comparative analysis (Section 4.2) shows that GMM-TS consistently outperforms state-of-the-art uni-modal and multi-modal baselines across a variety of domains. We also highlight its unique advantages (Section 4.3), including its ability to fuse more than two experts, support for efficient offline pretraining, and inherent interpretability through expert weighting (further discussed in Appendix D.3).

**Ablations:** we conduct ablations (Section 4.4) to validate key architectural choices: the aggregation method and adaptive approach of our gating module and the dimensionality of the shared latent space. These experiments collectively confirm GMM-TS's robust design and versatility.

### 4.1 EXPERIMENTAL SETUP

We follow the experimental protocol established in prior TSF work Wu et al. (2021b); Zhou et al. (2022c); Liu et al. (2023), and adopt the benchmark design and preprocessing pipeline from TimeMMD Liu et al. (2024a). Experiments are conducted across all nine domains in the TimeMMD benchmark, each containing aligned numerical and textual time series. The goal is to forecast future numerical values using multimodal historical inputs.

**Forecasting setup.** Forecast horizons are chosen following standard TSF settings Wu et al. (2021a); Zhou et al. (2022b); Nie et al. (2022a). For **daily datasets**, we use horizons {48, 96, 192, 336} with a lookback window of 96 and label length of 48. For **weekly datasets**, the horizons are {12, 24, 36, 48}, with lookback 36 and label length 18. For **monthly datasets**, we use {6, 8, 10, 12}, with lookback 8 and label length 4. Unless stated otherwise, we report results averaged across these horizons.

Each dataset consists of aligned numerical and textual time series. The textual signals vary in structure and semantics across domains: for instance, domains such as `Security`, `Traffic`, and `Energy` include structured log-style reports or alerts, while others like `Economy`, `Social Good`, and `Public Health` contain more descriptive event narratives or policy summaries. For detailed dataset statistics (rows, features per modality), see Appendix B.

**Expert models.** We evaluate five numerical forecasting models as TSF-N experts: **Transformer-based:** Reformer Kitaev et al. (2020), Informer Zhou et al. (2021), PatchTST Nie et al. (2022b); **MLP-based:** DLinear Zeng et al. (2023); and **architecture-agnostic:** FiLM Zhou et al. (2022a). These were selected to reflect a diversity of modeling strategies—ranging from lightweight MLPs to expressive attention-based architectures—providing complementary inductive biases and performance characteristics. For textual inputs, we use three pretrained LLMs—GPT-3.5, GPT-2 Small, and LLaMA-2—as TSF-T experts, chosen for their varied scale and encoder capacity. All LLM backbones are frozen during training, with only a task-specific MLP head fine-tuned, as described in Section 3. Prompt design follows the TimeMMD protocol to ensure consistency.

**Implementation details.** We optimize all models using Adam with early stopping for a maximum of 10 training epochs. The shared latent dimension of the gating module is set to $d_m = 256$. We apply early stopping on the validation loss, and empirically observe that all models converge within 10 epochs across all datasets. We provide additional details in Appendix C.

## 4.2 Comparative analysis of multi-modal time series forecasting methods

**Baseline comparative analysis**   We evaluate our framework across all 15 combinations of TSF-N and TSF-T expert pairs (5 numerical × 3 textual models), tested across 4 forecasting horizons on each of the 9 TimeMMD domains, resulting in 540 experiments. The main multimodal baselines are **TimeMMD** Liu et al. (2024a) and **GPT4MTS** Jia et al. (2024). TimeMMD is structurally closer to our method, combining frozen LLMs with forecasting backbones via a fusion mechanism. In contrast, GPT4MTS employs a monolithic LLM and does not allow configuration of component models. For completeness, we evaluate GPT4MTS on the same datasets and horizons. We also include the standalone performance of each unimodal model to contextualize their contributions when used in multimodal combinations. Table 1 compares the mean squared error (MSE) of GMM-TS, GPT4MTS, TimeMMD, and a unimodal baseline, averaged across all forecast horizons. To ensure a fair comparison with GPT4MTS, we use the same numerical and textual backbones (PatchTST and GPT2) across all systems. GMM-TS achieves the best or comparable performance across all domains, outperforming both baselines in 8 out of 9 domains.

Table 1: Domain-wise forecasting error for different methods. Values are *aggregated domain-level averages* over all forecast horizons and relevant expert combinations. The lowest value per domain is highlighted in bold. Detailed results per-domain, per-horizon, per-expert are provided in Tables 8-16 for multi-modal (pairwise) experiments and in Tables 17-18 for uni-modal experiments (Appendix D.1 and D.2)

| Domain | Unimodal | GPT4MTS | TimeMMD | GMM-TS |
|---|---|---|---|---|
| Agriculture | 0.10 | 0.23 | 0.11 | **0.09** |
| Climate | 1.32 | 1.27 | 1.15 | **1.02** |
| Economy | **0.02** | **0.02** | 0.04 | **0.02** |
| Energy | 0.28 | 0.28 | 0.29 | **0.27** |
| Environment | 0.52 | 0.59 | 0.47 | **0.41** |
| Public Health | 1.61 | 1.96 | 1.46 | **1.17** |
| Security | 116.43 | **74.91** | 112.90 | 110.30 |
| Social Good | 1.14 | 1.13 | 0.99 | **0.95** |
| Traffic | 0.21 | 0.22 | 0.20 | **0.19** |

**Comparative analysis across expert pairs.**   To further evaluate our gating mechanism, we conduct a detailed comparison with TimeMMD—the only baseline that supports configurable expert pairings. GPT4MTS is excluded from this analysis, as it does not support expert replacement. For each domain, forecast horizon, and expert pair, we compute the MSE for both GMM-TS and TimeMMD and define the performance gap as $\Delta = \text{MSE}_{\text{GMM-TS}} - \text{MSE}_{\text{TimeMMD}}$.

Table 2 shows, for each domain, the number of expert-horizon combinations (out of 60) in which each method achieves a lower MSE. GMM-TS outperforms TimeMMD in the vast majority of set-

tings. Table 3 further summarizes the distribution of $\Delta$ values across all expert pairs, reporting the mean and standard deviation, and percentage of pairs where GMM-TS performs better. GMM-TS achieves lower MSE in all nine domains with notable gains ($\geq$ 5% and up to 50%) for seven domains. These results highlight the effectiveness of our architecture in dynamically leveraging complementary signals from heterogeneous experts as well as the robustness and generality of our gating mechanism across domains and expert pairings.

Table 2: Comparison of TimeMMD and our method (GMM-TS) across 540 experiments spanning 9 domains. Each domain includes 60 expert pair evaluations (5 TSF-N × 3 TSF-T) across 4 forecasting horizons. The table reports, for each domain, how many times each method achieves a lower MSE (a 'win'). The higher count per domain is highlighted in bold.

| Domain | TimeMMD 'Wins' | GMM-TS 'Wins' (Ours) |
|---|---|---|
| Agriculture | 9 | **51** |
| Climate | 4 | **56** |
| Economy | 5 | **55** |
| Energy | 10 | **50** |
| Environment | 8 | **52** |
| Public Health | 4 | **56** |
| Security | 11 | **49** |
| Social Good | 25 | **35** |
| Traffic | 2 | **58** |

Table 3: Domain-wise comparison of GMM-TS and TimeMMD across expert pairs. Negative values of $\Delta$ Mean indicate a reduction in forecasting error by GMM-TS (i.e., improvement over TimeMMD). % Better reports the percentage of improvement.

| Domain | $\Delta$ Mean ($\downarrow$) | Std. Dev. | % Better ($\uparrow$) |
|---|---|---|---|
| Agriculture | -0.02 | 0.00 | 18.18% |
| Climate | -0.08 | 0.02 | 11.3% |
| Economy | -0.02 | 0.00 | 50.0% |
| Energy | -0.02 | 0.00 | 6.9% |
| Environment | -0.06 | -0.02 | 12.77% |
| Public Health | -0.29 | -0.18 | 19.86% |
| Security | -2.60 | -0.08 | 2.3% |
| Social Good | -0.04 | -0.02 | 4.04% |
| Traffic | -0.01 | -0.01 | 5.0% |

## 4.3 Additional benefits of GMM-TS

**Beyond expert pairs.** While prior work on multi-modal TSF Liu et al. (2024a); Jia et al. (2024) focuses on fusing a single TSF-N and TSF-T expert, GMM-TS supports flexible fusion over arbitrary sets of uni-modal experts. To demonstrate this capability, we evaluate triplet combinations of TSF-N and TSF-T models and compare their performance to the constituent expert pairs. Specifically, we compare the pairs {GPT3.5, DLinear}, {GPT3.5, Informer}, and {GPT3.5, PatchTST} to triplets such as {GPT3.5, DLinear, Informer}, and so on. Table 4 reports the average MSE across each configuration. In most domains, triplet combinations outperform their pairwise baselines, highlighting the advantage of fusing complementary signals from multiple modalities. Extended results, including expert quartets, are provided in Appendix D.3.1.

**Joint training versus offline pre-training.** GMM-TS can be adapted for offline pretraining of individual modality experts. Unlike the joint training approach (Section 3.2.2), this offline strategy separates expert learning from gating network training. Each expert is trained independently, and then their fixed outputs are combined by the gating module in a separate training phase. This modular approach offers flexibility and efficiency, allowing experts to be reused across different tasks without retraining. Figure 2 illustrates the additional training time for pairs as the number of experts per modality increases for TimeMMD and GMM-TS (both joint and offline). The curves for GMM-TS (offline/joint) are also annotated with the average percentage reduction in Mean Squared Error (MSE) across all domains, expert pairs, and prediction horizons. GMM-TS with offline pretraining

Table 4: Forecasting MSE achieved by our method (GMM-TS) when using expert pairs versus expert triplets. For each domain, we report the average MSE across all evaluated pair and triplet configurations. Lower values indicate better performance.

| Domain | Pairs MSE | Triplets MSE |
|---|---|---|
| Agriculture | 0.19 | **0.16** |
| Climate | **1.00** | 1.02 |
| Economy | 0.20 | **0.08** |
| Energy | 0.34 | **0.30** |
| Public Health | 1.27 | **1.21** |
| Security | 115.06 | **112.81** |
| Social Good | 0.94 | **0.90** |
| Traffic | 0.19 | **0.18** |

scales better with more experts and achieves comparable performance to TimeMMD. Joint training yields state-of-the-art MSE and requires less optimization time than TimeMMD.

**Interpretable multi-modal time series forecasting** The expert weight matrix (W) in GMM-TS provides *inherent interpretability* by showing the contribution of each expert at each forecast time step, revealing how the model combines information from different modalities. Fig. 3 illustrate two representative examples from the `Climate` (Fig. 3 - right) and `Social Good` (Fig. 3 - left) domains, respectively. In each figure, we plot the individual predictions of two experts (PatchTST in blue and GPT3.5 in orange), the fused output of GMM-TS (in green), and the ground truth (in black). We annotate each expert prediction with its corresponding gating weight ($w$), predicted by the Transformer-based gating network.

In the `Social Good` example (Fig. 3, right), GMM-TS assigns consistently higher weights to PatchTST, which closely tracks the ground truth. GPT3.5 receives lower weights throughout, particularly where its predictions diverge. This illustrates how the model prioritizes the more accurate modality at each time step. In the `Climate` example (Fig. 3, right), the gating network initially favors GPT3.5, whose early predictions are better aligned with the target. However, as the textual expert's accuracy degrades later in the sequence, the model shifts weight toward PatchTST, improving the overall forecast. These examples demonstrate how GMM-TS adapts its fusion strategy based on the changing accuracy of each expert over time, prioritizing the more reliable modalities. This provides an interpretable way to see the relative contribution of each expert.

Table 5: Ablation results for aggregation strategy (left) and gating dimension (right). Left: average MSE across three TSF-N expert combinations— {DLinear, Informer, Reformer}, {Informer, PatchTST, Reformer}, and {DLinear, Informer, PatchTST, Reformer}— under Direct, Hierarchical, and Latent aggregation. Right: average MSE for five expert pairs (GPT3.5 + one TSF-N model from {DLinear, FiLM, Informer, PatchTST, Reformer}) across gating dimensions $d_m \in \{32, 64, 128, 256, 512\}$. Lower is better.

| Domain | Agg. Strategy | | | Gating Dimension | | | | |
|---|---|---|---|---|---|---|---|---|
| | Direct | Hierarchical | Latent | 32 | 64 | 128 | 256 | 512 |
| Economy | **0.17** | 0.21 | 1.23 | 0.19 | **0.18** | 0.20 | **0.18** | 0.20 |
| Energy | 0.31 | 0.31 | **0.30** | 0.39 | **0.36** | **0.36** | **0.36** | **0.36** |
| Public Health | 1.24 | **1.21** | 1.29 | 1.30 | 1.28 | **1.27** | **1.27** | **1.27** |
| Security | **115.62** | 116.68 | 127.79 | **113.81** | 114.59 | 114.61 | 115.58 | 115.25 |
| Social Good | **0.85** | 0.87 | **0.85** | 1.00 | 0.98 | **0.96** | **0.96** | **0.96** |
| Traffic | **0.17** | **0.17** | 0.18 | **0.19** | **0.19** | **0.19** | **0.19** | **0.19** |

## 4.4 ABLATIONS

We investigate the effect of two key architectural components of GMM-TS: the strategy used to aggregate expert outputs, and the dimensionality of the shared latent space in the gating module.

**Aggregation strategy.** Our default approach, **Direct Aggregation**, combines expert predictions in the target space using a learned weight matrix applied per time step. We compare this to two alternatives (described in detail in Appendix E): (1) **Hierarchical Aggregation**, which first aggregates predictions within each modality before fusing them across modalities; and (2) **Latent Aggregation**, which aggregates the experts' latent representations and applies a projection head to produce the forecast.

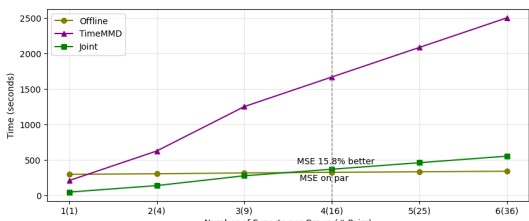

Figure 2: Overall training times for training expert pairs for TimeMMD and GMM-TS. We show the additional training time required when adding more experts per modality.

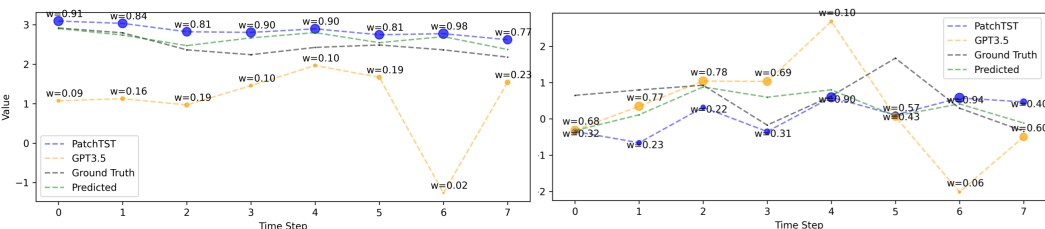

Figure 3: Visualization of expert contributions for the `Climate` (left) and `Social Good` (right) domains. The plots show predictions from PatchTST (blue) and GPT3.5 (orange), their per-time-step weights ($w$), the fused prediction (green), and the ground truth (black).

Table 5 (left) reports the average MSE across six domains for three multi-expert configurations. Direct aggregation consistently performed the best, achieving the lowest error in four out of six domains. Hierarchical aggregation performed similarly, suggesting that the primary benefit comes from fusing modalities rather than the specific fusion method. Latent aggregation proved less robust, with degraded performance in several domains, underscoring the value of directly combining expert predictions in the output space. Additional configurations are evaluated in Appendix F.

**Gating dimension.** We also evaluate the effect of varying the gating dimension $d_m$, which determines the shared latent space size in which expert representations are projected and fused. Table 5 (right) shows average MSE values across five expert pairs—formed by combining GPT-3.5 with each of DLinear, FiLM, Informer, PatchTST, and Reformer—under different settings of $d_m \in \{32, 64, 128, 256, 512\}$. The results are stable across settings, indicating that GMM-TS is robust to the choice of gating dimension. Additional results are reported in Appendix F.

**Adaptive versus static gating.** To evaluate the value of dynamic gating, we compared our adaptive gating network to a static, fixed-weight baseline, where a gating weight matrix is learned during training but fixed across all inputs at inference time.. As shown in Table 26 (Appendix F), the adaptive model significantly outperformed the static one across all domains, with a degradation of over 12x in the `Economy` domain without adaptive gating. These results highlight that input-conditioned gating is crucial for strong performance in diverse time series forecasting scenarios.

## 5 CONCLUSION

We introduced GMM-TS, a novel gating-based architecture for exogenous multi-modal time series forecasting that adaptively fuses predictions from multiple experts. To the best of our knowledge, it is the first approach to extend multi-modal TSF beyond expert pairs, enabling the integration of multiple specialists across modalities. GMM-TS consistently outperforms state-of-the-art baselines across diverse domains and expert configurations, and its gating mechanism provides built-in per-timestep interpretability by showing each expert's contribution.

**Limitations and Future Work** While we focused on numerical and textual inputs, GMM-TS can be extended to other modalities like visual time series data. Future work will explore these extensions, as well as other time series tasks like classification and anomaly detection. We also plan to investigate efficient adaptation techniques such as Low-Rank Adaptation (LoRA).

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

APPENDIX

We provide additional results and information to complement our main text:

- Section A details and compares related work.

- Section B provides additional details on the datasets used.

- Section C gives additional implementation details.

- Section D provided extended results:

  - Section D.1 provides per-domain, per-horizon, per expert-pair comparison between GMM-TS and TimeMMd

  - Section D.2 reports per-domain, per-horizon, per-expert unimodal results

  - Section D.3 provides extended results and discussion on additional benefits of our proposed method:

    * Section D.3.1 provides extended results for the experiments with more than two experts for multi-modal TSF.

    * Section D.3.2 provides extended results for evaluating the offline pre-training strategy.

    * Section D.3.3 discusses additional benefits not covered in the main text.

- Section E provides the formulation of the Hierarchical and Latent aggregation methods.

- Section F provides extended and additional ablation results.

- Section G discussed the broader impacts of our work.

## A    EXTENDED RELATED WORK AND COMPARISON

**Extended Discussion.**    In this section, we expand on prior work in both uni-modal and multi-modal time series forecasting, including recent MoE-based architectures.

- **TimeMMD** Liu et al. (2024a): Introduced the first large-scale multi-modal TSF benchmark. Its architecture supports modular expert inputs but fuses predictions via a fixed, global weight—resulting in a lack of dynamic, input-aware fusion.

- **GPT4MTS** (Jia et al., 2024): Trains a decoder-only GPT2 with soft prompts for modality-specific conditioning. Fusion is implicit in the prompt encoding, and the architecture lacks modularity.

- **Google-TSF** (Daswani et al., 2024): Adapts VLMs to combine plots with structured data for forecasting. Fusion is performed within a monolithic foundation model, lacking interpretability or explicit per-modality contributions.

- **Time-VLM** (Zhong et al., 2025): Uses tri-modal inputs (visualized plots, language context, and numerical sequences), but relies on vision-language models with concatenation-based fusion. This method does not offer expert modularity or explicit attribution.

- **MERA** (Zhou et al., 2025): Proposes a retrieval-augmented MoE framework for stock forecasting. While modular and scalable, MERA is limited to numerical inputs and does not support dynamic multi-modal interaction.

Table 6 compares GMM-TS with prior work. This comparison highlights that GMM-TS is the first method to combine dynamic, interpretable, and modular fusion in a multi-expert, multi-modal TSF setting.

To the best of our knowledge, GMM-TS is the first architecture for multi-modal TSF that enables adaptive, interpretable, and modular fusion over an arbitrary number of heterogeneous experts. It departs from prior methods by going beyond two-modality assumptions, offering plug-and-play flexibility, and supporting both joint and offline training regimes.

Table 6: Comparison of our method (GMM-TS) to prior work across key properties.

| Model | Modality | Fusion Type | Dynamic | Interpretable | Modular | # Experts |
|---|---|---|---|---|---|---|
| TimeMMD (Liu et al., 2024a) | Text + Num | Prediction (fixed weight) | ✗ | ✗ | ✓ | 2 |
| GPT4MTS (Jia et al., 2024) | Text + Num | Prompted representation | ✓ | ✗ | ✗ | 2 |
| Google-TSF (Daswani et al., 2024) | Vision + Num | Foundation model | ✓ | ✗ | ✗ | 2 |
| Time-VLM (Zhong et al., 2025) | Vision + Text + Num | Concatenated inputs | ✓ | ✗ | ✗ | 3 |
| MERA (Zhou et al., 2025) | Num | Expert + retrieval | ✓ | ✗ | ✓ | > 2 |
| **GMM-TS (Ours)** | Text + Num (+ more) | Prediction (learned weights) | ✓ | ✓ | ✓ | ≥ 2 |

## B  DATASET DETAILS

We conduct extensive evaluation experiments across a wide range of domains available from the TimeMMD benchmark. The domains included in this benchmark cover a range of real-world forecasting scenarios:

- **Agriculture**: Agricultural production and supply chain time series.
- **Climate**: Temperature, rainfall, and atmospheric readings with textual context.
- **Economy**: Financial and economic indicators tied to news reports.
- **Energy**: Power consumption and energy production data.
- **Environment**: Environmental monitoring data with text from reports.
- **Public**: Public health metrics (e.g., infection rates, hospitalizations).
- **Security**: Geopolitical/security event data with incident summaries.
- **Social Good**: Time series from social impact domains (e.g., education, inequality).
- **Traffic**: Vehicle usage and congestion statistics with urban planning documents.

Table 7 further summarizes the size and modality composition of each dataset in the TimeMMD benchmark (Liu et al., 2024a). Each domain consists of aligned textual and numerical time series across daily, weekly, or monthly frequencies.

Table 7: Dataset statistics used in our experiments. Each domain includes aligned numerical and textual features, with a single forecasting target (table reproduced from Liu et al. (2024b))

| Domain | # Timestamps | Dimensions |
|---|---|---|
| Agriculture | 496 | 1 |
| Climate | 496 | 5 |
| Economy | 423 | 3 |
| Energy | 1479 | 9 |
| Environment | 11102 | 4 |
| Public Health | 1389 | 11 |
| Security | 297 | 1 |
| Social Good | 900 | 1 |
| Traffic | 531 | 1 |

## C  IMPLEMENTATION DETAILS

Our multimodal time series forecasting system employs a comprehensive optimization strategy where all models are optimized using the Adam optimizer with early stopping configured for a maximum of 10 training epochs. The system utilizes multiple Adam optimizers with differentiated learning rates: time series models use a learning rate of 0.0001, MLP components use 1e-2, projection layers use 1e-3, and the gating module uses 1e-3. Early stopping is implemented with a patience of 5 epochs, monitoring validation loss across all model components including time series models, MLP layers, and the gating module. The shared latent dimension of the gating module 256 by default, though ablation studies systematically test values of 32, 64, 128, and 512 to evaluate the impact of gating dimension on performance.

The gating module architecture consists of an input embedding layer with time features, expert projection layers that map expert outputs to a shared dimension, a 2-layer transformer encoder with 8 attention heads, gating networks implemented as MLP layers for computing attention weights, and a final output projection layer. Training is conducted with a batch size of 32, MSE loss function, and optional mixed precision training, while the gating mechanism supports both "latent" and "prediction" input types for expert integration. The transformer encoder employs GELU activation, 0.1 dropout rate, and a feed-forward dimension of 2048, providing a robust framework for multimodal time series forecasting with systematic evaluation of different architectural configurations.

## D EXTENDED QUALITY RESULTS

### D.1 PAIRWISE COMPARISON WITH TIMEMMD BY DOMAIN

We present the *detailed per-horizon, per-expert pairwise results* that underlie the *aggregated domain-level averages* reported in Table 1 in the main text. In other words, Table 1 is a compact summary of the average performance across all forecast horizons (12, 24, 36, 48) and expert combinations for each domain, while the results shown here provide the full breakdown for transparency and reproducibility.

For each setting, we experiment with various combinations of numerical (TSF-N) and textual (TSF-T) experts, including both strong backbone models and LLM-based textual predictors. For each combination, we report the mean squared error (MSE) for GMM-TS (our proposed method) and for TimeMMD, using the same expert configuration. The lower MSE value in each row is highlighted in bold.

Our method consistently outperforms TimeMMD in terms of forecasting accuracy, as measured by MSE. Notably, this superiority holds for every domain, underscoring the robustness and generality of our gating-based fusion approach. These results demonstrate that our system not only offers improved performance but also scales reliably across different forecasting scenarios.

### D.2 UNI-MODAL EXPERIMENTS

Tables 17 and 18 present the *detailed per-horizon, per-expert* unimodal forecasting results for the `Energy` and `Public Health` domains, respectively, using weekly-resolution data. These results complement the *aggregated domain-level averages* reported in Table 1 in the main text, where results are averaged across multiple horizons and expert combinations. Here, each row corresponds to a specific horizon and a specific TSF-N or TSF-T expert, providing full transparency and enabling reproduction of the aggregated results.

For each setting, we report multiple evaluation metrics — Mean Absolute Error (MAE), MSE, Root Mean Squared Error (RMSE), Mean Absolute Percentage Error (MAPE), and Mean Squared Percentage Error (MSPE) — across various prediction lengths ('Horizon') and expert models. Both Transformer-based and MLP-based TSF-N models, as well as large language model-based TSF-T experts, are evaluated.

Each individual TSF-N or TSF-T expert consistently underperforms compared to our fused model that integrates the same expert with a complementary modality. This highlights the advantage of multi-modal fusion: combining numerical and textual representations yields improved forecasting accuracy across all domains and settings.

### D.3 ADDITIONAL BENEFITS: EXTENDED RESULTS AND DISCUSSION

Our proposed architecture introduces not only improvements in forecasting accuracy, but also meaningful benefits in terms of model transparency, usability, and extensibility. Below, we discuss several key aspects of the system beyond raw performance and the benefits already demonstrated in the main text.

### D.3.1 Going beyond expert pairs for multi-modal TSF: additional results

We provide additional results for TSF with more than two TSF-N experts. Table 19 compares the average MSE across pairs and respective triplets and quartets of experts. Table 20 provides an extended version of Table 4 in the main text, comparing the forecasting accuracy (in terms of MSE) of pairs to the respective triplet, for each triplet. Triplets and quartets surpass expert pairs on average, demonstrating the advantage of our method, in supporting multi-expert (more than two) setting.

### D.3.2 Offline expert pre-training: additional results

Table 21 compares GMM-TS (offline and joint training) with TimeMMD across all domains. GMM-TS with offline pre-training remains competitive but presents a substantial reduction in runtime (as shown in the main text). GMM-TS with joint training, consistently outperforms Time-MMD while presenting competitive train time (as shown in the main text).

### D.3.3 Discussion on additional benefits

**Debugging and failure analysis.**  The explicit separation between expert predictions and the gating module allows for effective error analysis. When performance degrades, it is possible to isolate whether the issue stems from a specific expert or from the fusion logic. This decomposition provides a structured debugging pathway, facilitating targeted improvements (e.g., re-training only the underperforming expert).

**Rationale inspection and human-in-the-loop validation.**  By making the gating decisions transparent and traceable, the model allows users to inspect the *rationale* behind its predictions—e.g., whether it relied on textual evidence, numerical trends, or both. This is especially useful in high-stakes domains (e.g., healthcare, infrastructure monitoring), where human oversight is critical. Furthermore, by comparing the fused forecast against individual expert outputs, domain experts can evaluate when the model is being conservative, overconfident, or appropriately balanced.

**Guidance for few-shot expert fine-tuning.**  In scenarios where certain experts perform poorly (e.g., domain shift, low-resource domains), our architecture provides actionable feedback: if the gating module consistently down-weights a particular expert, it can serve as a signal to fine-tune that expert using a small number of task-specific examples. This opens the door to a principled, few-shot training loop, where human intervention is guided by model behavior rather than guesswork.

**Modularity and extensibility.**  Finally, our design is inherently modular. New experts—whether trained on different modalities, domains, or tasks—can be plugged into the system (when using offline pretraining this further means existing experts do not need to be re-trained). The gating module adapts to newly introduced signals by updating the expert weights accordingly. This makes our approach especially well-suited to evolving multi-modal pipelines in real-world applications.

## E  Hierarchical and Latent Aggregations

We provide the formulation of the Hierarchical and Latent aggregation methods below:

- **Hierarchical aggregation**: In this variant, we first derive two distinct weight matrices, $W_n \in \mathbb{R}^{p \times |E_n|}$ and $W_t \in \mathbb{R}^{p \times |E_t|}$ from the original weight matrix $W$. This derivation involves applying the Softmax function to the indices of the experts that belong to the numeric modality ($E_n$) and the textual modality ($E_t$)), respectively. Subsequently, we use these modality-specific weight matrices $W_n$ and $W_t$ in our Direct Aggregation method (Eq. 3) to predict the future time series based on each modality individually, resulting in $Y_n$ (numeric-based forecast) and $Y_t$ (textual-based forecast). To combine these uni-modal predictions, we predict a weight vector $w \in \mathbb{R}^p$, where each element corresponds to a time step in the forecast horizon. This weight vector is generated by applying an additional MLP head to to $g_o$. The final forecast Y is then computed as a weighted average of the uni-modal predictions:

$$Y = (1 - w) \cdot Y_t + w \cdot Y_n \tag{5}$$

- **Latent aggregation**: In this variant, instead of directly aggregating the experts' predictions, we aggregate the projected *latent* representations of their predictions. Similar to our primary aggregation method (Section 3.2.1), we combine these latent features (given by $H_s$) into a fused latent vector $h$ using the weight matrix $W \in \mathbb{R}^{d^* \times |E|}$ with $d* = d_m$:

$$h = [\sum_{j=0}^{|E|-1} H_s[i,j]W[i,j]]_{i=0...d_m-1} \tag{6}$$

Subsequently, we employ an additional fully connected layer to regress $Y$ from the fused latent vector $h$.

## F  ABLATIONS: ADDITIONAL RESULTS

We provide extended ablation results, including additional ablation experiments.

Tables 22 and Table 24 provide the extended version of Table5 in the main text, comparing different aggregation methods and gating dimensions. Table 23 further compares the Direct and Latent aggregations across expert pairs.

### F.1  ABLATION STUDY: ADAPTIVE VERSUS STATIC WEIGHTS

To isolate the contribution of dynamic gating, we compare our adaptive transformer-based gating network with a static learned-weight baseline. In this ablation, the transformer module is replaced with a fixed weight matrix (per horizon and expert index), learned during training but fixed across all inputs at inference time. This setting removes adaptivity, forcing the model to rely on global average weights.

**Setup**  We evaluate across all benchmark domains using the same expert pool as in the main experiments (TSF-N: {DLinear, Informer, Reformer, PatchTST}, TSF-T: GPT-3.5).

**Experiment Motivation**  The goal of this ablation is to quantify the contribution of our dynamic gating mechanism. We ask: *What happens if, instead of using a transformer to compute input-dependent aggregation weights, we learn fixed weights during training and use them unchanged at inference?*

**Methodology   Regular (Dynamic) Method:**

- Uses a transformer encoder to dynamically compute gating weights at each step.
- Weights are conditioned on the current input context and expert latent representations.
- Adaptively shifts emphasis among experts at inference time.

**Ablation (Fixed) Method:**

- Replaces the transformer gating module with a learned static weight matrix $W$.
- $W \in \mathbb{R}^{\text{num\_experts} \times \text{prediction\_horizon}}$ is optimized during training and fixed thereafter.
- Same number of parameters in the fusion stage, but no dependence on input context during inference.

**Summary by Expert and Domain**  To better understand the sensitivity of different configurations to the removal of dynamic gating, we average results by TSF-N expert (across all domains) and by domain (across all experts).

**Key Insights**

- **Dynamic gating is essential:** Across all 143 configurations tested, removing dynamic gating increases MSE by an average of **328.3%**.

- **By Expert:** DLinear is the most robust TSF-N expert (55.9% avg. degradation), while Informer is the most sensitive (68.7% avg. degradation). PatchTST and Reformer are moderately sensitive ( 66–68%).

- **By Domain:** Economy is catastrophically sensitive (**1,272.8%** degradation on average), Energy is moderately impacted (114.6%), Traffic has substantial degradation (92.7%), SocialGood is relatively resilient (31.1%), and Security shows minimal impact (8.6%).

- **Notable cases:**
  - *Best case:* Security + DLinear shows a slight improvement when gating is removed (**-0.2%**).
  - *Worst case:* Economy + Informer suffers a catastrophic degradation (**3120.5%**).

- **Interpretation:**
  - Domains with high variability and strong cross-modal dependencies (e.g., Economy) rely heavily on adaptive expert weighting.
  - Stable experts (e.g., DLinear) in low-variability domains (e.g., Security) are less reliant on dynamic gating and may even be marginally unaffected.
  - Experts with more complex temporal modeling (e.g., Informer) appear to depend more on gating adaptivity to leverage cross-expert complementarity.

**Conclusion**   Dynamic gating is a critical component of the architecture. In some domains, replacing it with fixed weights reduces performance by over an order of magnitude. Even in domains with smaller gains, dynamic gating remains at least competitive and often substantially better.

**Full Results**   Table27-31 provide the complete per-domain, per-configuration results. Regular MSE refers to the dynamic gating variant; Ablation MSE refers to the fixed-weight variant. Boldface indicates the lower MSE in each row.

## F.2   ABLATION STUDY: NUMBER OF TRANSFORMER LAYERS IN THE GATING MODULE

**Motivation**   The gating module in GMM-TS uses a Transformer encoder to compute input-dependent weights over experts. While our main experiments use a lightweight 2-layer encoder for computational efficiency, the optimal depth for this component is not obvious. We therefore perform an ablation varying the number of encoder layers to examine how depth impacts performance.

**Why only Informer and PatchTST?**   To keep the analysis focused and interpretable, we report results for two representative TSF-N experts: (1) **Informer**, a strong Transformer-based forecasting model, and (2) **PatchTST**, a state-of-the-art patch-based Transformer forecaster. These were selected because they represent high-performing but architecturally distinct approaches, and they show different sensitivity patterns to gating depth. The TSF-T expert is fixed to GPT-3.5 for all experiments.

**Methodology**   We evaluate the gating module with **1, 2, 4, 6, and 8 encoder layers** across all benchmark domains. Some combinations were not run due to compute constraints, but the evaluation design is consistent for every domain. All other components and hyperparameters are fixed, including:

- Gating dimension $d_m = 64$
- Fixed aggregation type: direct
- Forecast horizons: multiple values per domain, but here we report selected key horizons

For each configuration, we report MSE and mark the lowest value in each row in **bold**.

**Per-domain results**   Tables 32–32 present the results for each domain. For Public Health domain, we only report the results for **pred_len=12** and **pred_len=24**, as these cover the most relevant forecasting scenarios in our benchmark.

**Average MSE across layers.** To complement the per-configuration results in Tables 32–34, we compute the average MSE for each number of Transformer encoder layers in the gating module, averaged across both Informer and PatchTST experts for each domain (Table 38). **Insights:**

- There is **no universal optimal number of layers**—performance varies by domain.
- In **Economy** and **Security**, **2 layers** achieve the lowest average error.
- In **Energy** and **Traffic**, deeper gating (6–8 layers) yields small improvements over shallower models.
- **SocialGood** benefits from moderately deep gating (4–8 layers), but differences are modest.
- Across all domains, **2–4 layers** offer a strong trade-off between accuracy and computational efficiency.

**Conclusion** While increasing the number of Transformer layers in the gating module can improve performance in some scenarios, the effect is domain- and model-dependent. A 2–4 layer configuration provides a robust, lightweight default that works well across both Informer and PatchTST without incurring excessive computational cost.

# G  BROADER IMPACTS

Multi-modal TSF, which integrates various data modalities like time series, text, images, and audio for prediction, can positively impact a wide range of domains, such as finance, healthcare, security, agriculture and more. Fusing signals from multiple modalities can yield more robust and accurate predictions, as also demonstrated by our work. Beyond improved accuracy, our work also provides intuitive interpretation, supporting informed decision making. As with any machine learning technology, the use of method and models should be done in a principled manner and for advancing the greater good.

Table 8: Pairwise MSE comparison for the `Agriculture` domain.

| Horizon | TSF-N Expert | TSF-T Expert | GMM-TS (MSE) | TimeMMD (MSE) | Delta |
|---|---|---|---|---|---|
| 6 | DLinear | GPT3.5 | **0.0798** | **0.0737** | 0.01 |
| 6 | DLinear | GPT2 | **0.0729** | 0.0738 | -0 |
| 6 | DLinear | LLAMA2 | **0.0736** | 0.0742 | -0 |
| 6 | FiLM | GPT3.5 | **0.0610** | 0.0896 | -0.03 |
| 6 | FiLM | GPT2 | **0.0617** | 0.0895 | -0.03 |
| 6 | FiLM | LLAMA2 | **0.0633** | 0.0926 | -0.03 |
| 6 | Informer | GPT3.5 | **0.3314** | 0.3378 | -0.01 |
| 6 | Informer | GPT2 | 0.3120 | **0.3013** | 0.01 |
| 6 | Informer | LLAMA2 | 0.2660 | **0.2493** | 0.02 |
| 6 | PatchTST | GPT3.5 | **0.0593** | 0.0775 | -0.02 |
| 6 | PatchTST | GPT2 | **0.0604** | 0.0845 | -0.02 |
| 6 | PatchTST | LLAMA2 | **0.0584** | 0.0733 | -0.01 |
| 6 | Reformer | GPT3.5 | **0.1801** | 0.2743 | -0.09 |
| 6 | Reformer | GPT2 | 0.3302 | **0.2268** | 0.1 |
| 6 | Reformer | LLAMA2 | 0.2108 | **0.1917** | 0.02 |
| 8 | DLinear | GPT3.5 | **0.1079** | 0.1871 | -0.08 |
| 8 | DLinear | GPT2 | **0.1114** | 0.1878 | -0.08 |
| 8 | DLinear | LLAMA2 | **0.1044** | 0.1824 | -0.08 |
| 8 | FiLM | GPT3.5 | **0.0792** | 0.1061 | -0.03 |
| 8 | FiLM | GPT2 | **0.0796** | 0.1067 | -0.03 |
| 8 | FiLM | LLAMA2 | **0.0837** | 0.1089 | -0.03 |
| 8 | Informer | GPT3.5 | **0.2043** | 0.3530 | -0.15 |
| 8 | Informer | GPT2 | **0.2201** | 0.3396 | -0.12 |
| 8 | Informer | LLAMA2 | 0.3107 | **0.2797** | 0.03 |
| 8 | PatchTST | GPT3.5 | **0.0761** | 0.0969 | -0.02 |
| 8 | PatchTST | GPT2 | **0.0764** | 0.0936 | -0.02 |
| 8 | PatchTST | LLAMA2 | **0.0778** | 0.0968 | -0.02 |
| 8 | Reformer | GPT3.5 | **0.2759** | 0.4320 | -0.16 |
| 8 | Reformer | GPT2 | 0.2515 | **0.2377** | 0.01 |
| 8 | Reformer | LLAMA2 | **0.1511** | 0.2542 | -0.1 |
| 10 | DLinear | GPT3.5 | **0.1258** | 0.1473 | -0.02 |
| 10 | DLinear | GPT2 | **0.1275** | 0.1471 | -0.02 |
| 10 | DLinear | LLAMA2 | **0.1145** | 0.1428 | -0.03 |
| 10 | FiLM | GPT3.5 | **0.1031** | 0.1352 | -0.03 |
| 10 | FiLM | GPT2 | **0.1033** | 0.1269 | -0.02 |
| 10 | FiLM | LLAMA2 | **0.1086** | 0.1261 | -0.02 |
| 10 | Informer | GPT3.5 | **0.3765** | 0.4100 | -0.03 |
| 10 | Informer | GPT2 | **0.3492** | 0.4090 | -0.06 |
| 10 | Informer | LLAMA2 | **0.2950** | 0.4360 | -0.14 |
| 10 | PatchTST | GPT3.5 | **0.0957** | 0.1159 | -0.02 |
| 10 | PatchTST | GPT2 | **0.0947** | 0.1251 | -0.03 |
| 10 | PatchTST | LLAMA2 | **0.0951** | 0.1084 | -0.01 |
| 10 | Reformer | GPT3.5 | **0.1761** | 0.3834 | -0.21 |
| 10 | Reformer | GPT2 | **0.2536** | 0.3265 | -0.07 |
| 10 | Reformer | LLAMA2 | 0.4337 | **0.4116** | 0.02 |
| 12 | DLinear | GPT3.5 | **0.1413** | 0.1626 | -0.02 |
| 12 | DLinear | GPT2 | **0.1442** | 0.1626 | -0.02 |
| 12 | DLinear | LLAMA2 | **0.1415** | 0.1543 | -0.01 |
| 12 | FiLM | GPT3.5 | **0.1240** | 0.1488 | -0.02 |
| 12 | FiLM | GPT2 | **0.1262** | 0.1494 | -0.02 |
| 12 | FiLM | LLAMA2 | **0.1238** | 0.1515 | -0.03 |
| 12 | Informer | GPT3.5 | **0.5398** | 0.5558 | -0.02 |
| 12 | Informer | GPT2 | **0.5124** | 0.5286 | -0.02 |
| 12 | Informer | LLAMA2 | **0.3751** | 0.6909 | -0.32 |
| 12 | PatchTST | GPT3.5 | **0.1222** | 0.1380 | -0.02 |
| 12 | PatchTST | GPT2 | **0.1227** | 0.1429 | -0.02 |
| 12 | PatchTST | LLAMA2 | **0.1254** | 0.1420 | -0.02 |
| 12 | Reformer | GPT3.5 | **0.2449** | 0.4353 | -0.19 |
| 12 | Reformer | GPT2 | **0.2729** | 0.5461 | -0.27 |
| 12 | Reformer | LLAMA2 | 0.4597 | **0.4058** | 0.05 |

Table 9: Pairwise MSE comparison for the `Climate` domain.

| Horizon | TSF-N Expert | TSF-T Expert | GMM-TS (MSE) | TimeMMD (MSE) | Delta |
|---|---|---|---|---|---|
| 6 | DLinear | GPT3.5 | **0.9548** | 1.1090 | -0.15 |
| 6 | DLinear | GPT2 | **1.0232** | 1.1090 | -0.09 |
| 6 | DLinear | LLAMA2 | **1.0461** | 1.1132 | -0.07 |
| 6 | FiLM | GPT3.5 | **1.0510** | 1.1845 | -0.13 |
| 6 | FiLM | GPT2 | **1.0474** | 1.1839 | -0.14 |
| 6 | FiLM | LLAMA2 | **1.0255** | 1.1833 | -0.16 |
| 6 | Informer | GPT3.5 | **0.9460** | 1.0851 | -0.14 |
| 6 | Informer | GPT2 | **0.9509** | 1.0549 | -0.1 |
| 6 | Informer | LLAMA2 | **0.9510** | 1.0562 | -0.11 |
| 6 | PatchTST | GPT3.5 | **0.9907** | 1.1205 | -0.13 |
| 6 | PatchTST | GPT2 | **1.0261** | 1.1467 | -0.12 |
| 6 | PatchTST | LLAMA2 | **1.0697** | 1.1652 | -0.1 |
| 6 | Reformer | GPT3.5 | **1.0126** | 1.2702 | -0.26 |
| 6 | Reformer | GPT2 | **1.0614** | 1.1178 | -0.06 |
| 6 | Reformer | LLAMA2 | 1.0796 | **1.0697** | 0.01 |
| 8 | DLinear | GPT3.5 | **0.9714** | 1.1505 | -0.18 |
| 8 | DLinear | GPT2 | **0.9458** | 1.1509 | -0.21 |
| 8 | DLinear | LLAMA2 | **0.9682** | 1.1394 | -0.17 |
| 8 | FiLM | GPT3.5 | **1.0319** | 1.1496 | -0.12 |
| 8 | FiLM | GPT2 | **1.0389** | 1.1483 | -0.11 |
| 8 | FiLM | LLAMA2 | **1.0065** | 1.1546 | -0.15 |
| 8 | Informer | GPT3.5 | **1.0439** | 1.0508 | -0.01 |
| 8 | Informer | GPT2 | **1.0271** | 1.0675 | -0.04 |
| 8 | Informer | LLAMA2 | **1.0528** | 1.0680 | -0.02 |
| 8 | PatchTST | GPT3.5 | **1.0390** | 1.1206 | -0.08 |
| 8 | PatchTST | GPT2 | **1.0222** | 1.1361 | -0.11 |
| 8 | PatchTST | LLAMA2 | **1.0215** | 1.1341 | -0.11 |
| 8 | Reformer | GPT3.5 | 1.0411 | **0.9818** | 0.06 |
| 8 | Reformer | GPT2 | 1.0881 | **1.0038** | 0.08 |
| 8 | Reformer | LLAMA2 | **1.0673** | 1.0739 | -0.01 |
| 10 | DLinear | GPT3.5 | **1.0050** | 1.1200 | -0.12 |
| 10 | DLinear | GPT2 | **1.0058** | 1.1212 | -0.12 |
| 10 | DLinear | LLAMA2 | **0.9708** | 1.1126 | -0.14 |
| 10 | FiLM | GPT3.5 | **0.9961** | 1.1458 | -0.15 |
| 10 | FiLM | GPT2 | **0.9962** | 1.1479 | -0.15 |
| 10 | FiLM | LLAMA2 | **1.0300** | 1.1525 | -0.12 |
| 10 | Informer | GPT3.5 | **1.0157** | 1.1208 | -0.11 |
| 10 | Informer | GPT2 | **0.9768** | 1.1127 | -0.14 |
| 10 | Informer | LLAMA2 | **1.0322** | 1.1316 | -0.1 |
| 10 | PatchTST | GPT3.5 | **1.0172** | 1.1678 | -0.15 |
| 10 | PatchTST | GPT2 | **1.0403** | 1.2201 | -0.18 |
| 10 | PatchTST | LLAMA2 | **1.0490** | 1.1373 | -0.09 |
| 10 | Reformer | GPT3.5 | **0.9511** | 1.0186 | -0.07 |
| 10 | Reformer | GPT2 | **0.9440** | 1.0205 | -0.08 |
| 10 | Reformer | LLAMA2 | 1.1014 | **1.0674** | 0.03 |
| 12 | DLinear | GPT3.5 | **0.9862** | 1.1171 | -0.13 |
| 12 | DLinear | GPT2 | **0.9955** | 1.1197 | -0.12 |
| 12 | DLinear | LLAMA2 | **0.9917** | 1.1229 | -0.13 |
| 12 | FiLM | GPT3.5 | **0.9864** | 1.1514 | -0.16 |
| 12 | FiLM | GPT2 | **1.0023** | 1.1516 | -0.15 |
| 12 | FiLM | LLAMA2 | **1.0154** | 1.1605 | -0.15 |
| 12 | Informer | GPT3.5 | **1.0128** | 1.1608 | -0.15 |
| 12 | Informer | GPT2 | **1.0246** | 1.1412 | -0.12 |
| 12 | Informer | LLAMA2 | **0.9887** | 1.0386 | -0.05 |
| 12 | PatchTST | GPT3.5 | **1.0182** | 1.1958 | -0.18 |
| 12 | PatchTST | GPT2 | **1.0416** | 1.1296 | -0.09 |
| 12 | PatchTST | LLAMA2 | **1.0364** | 1.1528 | -0.12 |
| 12 | Reformer | GPT3.5 | **0.9570** | 1.0451 | -0.09 |
| 12 | Reformer | GPT2 | **0.9899** | 1.0533 | -0.06 |
| 12 | Reformer | LLAMA2 | **0.9611** | 1.0479 | -0.09 |

Table 10: Pairwise MSE comparison for the `Economy` domain.

| Horizon | TSF-N Expert | TSF-T Expert | GMM-TS (MSE) | TimeMMD (MSE) | Delta |
|---|---|---|---|---|---|
| 6 | DLinear | GPT3.5 | 0.0461 | **0.0286** | 0.02 |
| 6 | DLinear | GPT2 | 0.0480 | **0.0286** | 0.02 |
| 6 | DLinear | LLAMA2 | 0.0316 | **0.0288** | 0 |
| 6 | FiLM | GPT3.5 | **0.0236** | 0.0507 | -0.03 |
| 6 | FiLM | GPT2 | **0.0241** | 0.0507 | -0.03 |
| 6 | FiLM | LLAMA2 | **0.0260** | 0.0496 | -0.02 |
| 6 | Informer | GPT3.5 | **0.3752** | 0.8354 | -0.46 |
| 6 | Informer | GPT2 | **0.3071** | 0.7657 | -0.46 |
| 6 | Informer | LLAMA2 | **0.5332** | 0.7260 | -0.19 |
| 6 | PatchTST | GPT3.5 | **0.0175** | 0.0370 | -0.02 |
| 6 | PatchTST | GPT2 | **0.0219** | 0.0399 | -0.02 |
| 6 | PatchTST | LLAMA2 | **0.0173** | 0.0430 | -0.03 |
| 6 | Reformer | GPT3.5 | **0.3330** | 0.7056 | -0.37 |
| 6 | Reformer | GPT2 | **0.1908** | 0.6518 | -0.46 |
| 6 | Reformer | LLAMA2 | **0.2272** | 0.3340 | -0.11 |
| 8 | DLinear | GPT3.5 | **0.0293** | 0.0850 | -0.06 |
| 8 | DLinear | GPT2 | **0.0199** | 0.0855 | -0.07 |
| 8 | DLinear | LLAMA2 | **0.0203** | 0.0793 | -0.06 |
| 8 | FiLM | GPT3.5 | **0.0199** | 0.0511 | -0.03 |
| 8 | FiLM | GPT2 | **0.0245** | 0.0517 | -0.03 |
| 8 | FiLM | LLAMA2 | **0.0270** | 0.0512 | -0.02 |
| 8 | Informer | GPT3.5 | **0.5743** | 0.8589 | -0.28 |
| 8 | Informer | GPT2 | **0.5804** | 0.8200 | -0.24 |
| 8 | Informer | LLAMA2 | **0.5187** | 1.1498 | -0.63 |
| 8 | PatchTST | GPT3.5 | **0.0168** | 0.0364 | -0.02 |
| 8 | PatchTST | GPT2 | **0.0158** | 0.0380 | -0.02 |
| 8 | PatchTST | LLAMA2 | **0.0260** | 0.0372 | -0.01 |
| 8 | Reformer | GPT3.5 | **0.3827** | 0.5075 | -0.12 |
| 8 | Reformer | GPT2 | 0.4512 | **0.4153** | 0.04 |
| 8 | Reformer | LLAMA2 | **0.2017** | 0.3871 | -0.19 |
| 10 | DLinear | GPT3.5 | **0.0334** | 0.0391 | -0.01 |
| 10 | DLinear | GPT2 | **0.0317** | 0.0391 | -0.01 |
| 10 | DLinear | LLAMA2 | **0.0258** | 0.0369 | -0.01 |
| 10 | FiLM | GPT3.5 | **0.0339** | 0.0511 | -0.02 |
| 10 | FiLM | GPT2 | **0.0337** | 0.0510 | -0.02 |
| 10 | FiLM | LLAMA2 | **0.0201** | 0.0523 | -0.03 |
| 10 | Informer | GPT3.5 | **0.7035** | 0.9927 | -0.29 |
| 10 | Informer | GPT2 | **0.6983** | 0.9676 | -0.27 |
| 10 | Informer | LLAMA2 | **0.5560** | 0.9261 | -0.37 |
| 10 | PatchTST | GPT3.5 | **0.0162** | 0.0384 | -0.02 |
| 10 | PatchTST | GPT2 | **0.0228** | 0.0386 | -0.02 |
| 10 | PatchTST | LLAMA2 | **0.0187** | 0.0382 | -0.02 |
| 10 | Reformer | GPT3.5 | **0.2334** | 0.2881 | -0.05 |
| 10 | Reformer | GPT2 | 0.1975 | **0.1632** | 0.03 |
| 10 | Reformer | LLAMA2 | **0.4508** | 0.8762 | -0.43 |
| 12 | DLinear | GPT3.5 | **0.0202** | 0.0294 | -0.01 |
| 12 | DLinear | GPT2 | **0.0213** | 0.0295 | -0.01 |
| 12 | DLinear | LLAMA2 | **0.0233** | 0.0282 | -0 |
| 12 | FiLM | GPT3.5 | **0.0175** | 0.0507 | -0.03 |
| 12 | FiLM | GPT2 | **0.0175** | 0.0507 | -0.03 |
| 12 | FiLM | LLAMA2 | **0.0222** | 0.0509 | -0.03 |
| 12 | Informer | GPT3.5 | **0.5856** | 1.0744 | -0.49 |
| 12 | Informer | GPT2 | **0.5823** | 1.0777 | -0.5 |
| 12 | Informer | LLAMA2 | **0.5828** | 0.9330 | -0.35 |
| 12 | PatchTST | GPT3.5 | **0.0286** | 0.0357 | -0.01 |
| 12 | PatchTST | GPT2 | **0.0289** | 0.0361 | -0.01 |
| 12 | PatchTST | LLAMA2 | **0.0244** | 0.0380 | -0.01 |
| 12 | Reformer | GPT3.5 | **0.1852** | 0.2172 | -0.03 |
| 12 | Reformer | GPT2 | **0.1729** | 0.3333 | -0.16 |
| 12 | Reformer | LLAMA2 | **0.3527** | 0.8047 | -0.45 |

Table 11: Pairwise MSE comparison for the `Energy` domain.

| Horizon | TSF-N Expert | TSF-T Expert | GMM-TS (MSE) | TimeMMD (MSE) | Delta |
|---|---|---|---|---|---|
| 12 | DLinear | GPT3.5 | **0.2291** | 0.2492 | -0.02 |
| 12 | DLinear | GPT2 | **0.2298** | 0.2491 | -0.02 |
| 12 | DLinear | LLAMA2 | **0.2371** | 0.2514 | -0.01 |
| 12 | FiLM | GPT3.5 | **0.2121** | 0.2451 | -0.03 |
| 12 | FiLM | GPT2 | **0.2126** | 0.2451 | -0.03 |
| 12 | FiLM | LLAMA2 | **0.2185** | 0.2449 | -0.03 |
| 12 | Informer | GPT3.5 | 0.2860 | **0.1679** | 0.12 |
| 12 | Informer | GPT2 | 0.3118 | **0.1701** | 0.14 |
| 12 | Informer | LLAMA2 | 0.2181 | **0.2118** | 0.01 |
| 12 | PatchTST | GPT3.5 | **0.1076** | 0.1267 | -0.02 |
| 12 | PatchTST | GPT2 | **0.1076** | 0.1268 | -0.02 |
| 12 | PatchTST | LLAMA2 | **0.1125** | 0.1313 | -0.02 |
| 12 | Reformer | GPT3.5 | **0.1962** | 0.3193 | -0.12 |
| 12 | Reformer | GPT2 | **0.3125** | 0.3466 | -0.03 |
| 12 | Reformer | LLAMA2 | **0.3191** | 0.3341 | -0.02 |
| 24 | DLinear | GPT3.5 | **0.3038** | 0.3507 | -0.05 |
| 24 | DLinear | GPT2 | **0.3054** | 0.3477 | -0.04 |
| 24 | DLinear | LLAMA2 | **0.3108** | 0.3481 | -0.04 |
| 24 | FiLM | GPT3.5 | **0.2872** | 0.3282 | -0.04 |
| 24 | FiLM | GPT2 | **0.2881** | 0.3326 | -0.04 |
| 24 | FiLM | LLAMA2 | **0.3071** | 0.3404 | -0.03 |
| 24 | Informer | GPT3.5 | 0.3591 | **0.3068** | 0.05 |
| 24 | Informer | GPT2 | 0.3370 | **0.2972** | 0.04 |
| 24 | Informer | LLAMA2 | **0.2934** | 0.3149 | -0.02 |
| 24 | PatchTST | GPT3.5 | **0.2299** | 0.2424 | -0.01 |
| 24 | PatchTST | GPT2 | **0.2297** | 0.2424 | -0.01 |
| 24 | PatchTST | LLAMA2 | 0.2366 | **0.2337** | 0 |
| 24 | Reformer | GPT3.5 | 0.4544 | **0.4536** | 0 |
| 24 | Reformer | GPT2 | **0.4454** | 0.4552 | -0.01 |
| 24 | Reformer | LLAMA2 | **0.4729** | 0.4873 | -0.01 |
| 36 | DLinear | GPT3.5 | **0.3767** | 0.4049 | -0.03 |
| 36 | DLinear | GPT2 | **0.3729** | 0.4046 | -0.03 |
| 36 | DLinear | LLAMA2 | **0.3873** | 0.4186 | -0.03 |
| 36 | FiLM | GPT3.5 | **0.3911** | 0.4439 | -0.05 |
| 36 | FiLM | GPT2 | **0.4004** | 0.4436 | -0.04 |
| 36 | FiLM | LLAMA2 | **0.3882** | 0.4498 | -0.06 |
| 36 | Informer | GPT3.5 | **0.4189** | 0.4740 | -0.06 |
| 36 | Informer | GPT2 | **0.4659** | 0.4765 | -0.01 |
| 36 | Informer | LLAMA2 | **0.3991** | 0.4227 | -0.02 |
| 36 | PatchTST | GPT3.5 | **0.3364** | 0.3490 | -0.01 |
| 36 | PatchTST | GPT2 | **0.3357** | 0.3489 | -0.01 |
| 36 | PatchTST | LLAMA2 | 0.3326 | **0.3237** | 0.01 |
| 36 | Reformer | GPT3.5 | **0.4998** | 0.5570 | -0.06 |
| 36 | Reformer | GPT2 | **0.5200** | 0.5558 | -0.04 |
| 36 | Reformer | LLAMA2 | **0.4315** | 0.5670 | -0.14 |
| 48 | DLinear | GPT3.5 | **0.4931** | 0.5236 | -0.03 |
| 48 | DLinear | GPT2 | **0.4927** | 0.5236 | -0.03 |
| 48 | DLinear | LLAMA2 | **0.4854** | 0.5267 | -0.04 |
| 48 | FiLM | GPT3.5 | **0.5007** | 0.5816 | -0.08 |
| 48 | FiLM | GPT2 | **0.4985** | 0.5800 | -0.08 |
| 48 | FiLM | LLAMA2 | **0.5038** | 0.5925 | -0.09 |
| 48 | Informer | GPT3.5 | 0.5858 | **0.5364** | 0.05 |
| 48 | Informer | GPT2 | **0.4660** | 0.5319 | -0.07 |
| 48 | Informer | LLAMA2 | 0.5547 | **0.5430** | 0.01 |
| 48 | PatchTST | GPT3.5 | **0.4026** | 0.4388 | -0.04 |
| 48 | PatchTST | GPT2 | **0.4032** | 0.4401 | -0.04 |
| 48 | PatchTST | LLAMA2 | **0.4265** | 0.4370 | -0.01 |
| 48 | Reformer | GPT3.5 | **0.5577** | 0.5950 | -0.04 |
| 48 | Reformer | GPT2 | **0.5454** | 0.5571 | -0.01 |
| 48 | Reformer | LLAMA2 | **0.5363** | 0.5888 | -0.05 |

Table 12: Pairwise MSE comparison for the `Environment` domain.

| Horizon | TSF-N Expert | TSF-T Expert | GMM-TS (MSE) | TimeMMD (MSE) | Delta |
|---|---|---|---|---|---|
| 48 | DLinear | BERT | 0.41 | **0.46** | -0.04 |
| 48 | DLinear | GPT2 | 0.41 | **0.46** | -0.05 |
| 48 | DLinear | LLAMA2 | 0.41 | **0.46** | -0.04 |
| 48 | FiLM | BERT | 0.41 | **0.46** | -0.05 |
| 48 | FiLM | GPT2 | 0.41 | **0.46** | -0.04 |
| 48 | FiLM | LLAMA2 | 0.41 | **0.46** | -0.05 |
| 48 | Informer | BERT | 0.45 | **0.49** | -0.04 |
| 48 | Informer | GPT2 | 0.45 | **0.48** | -0.03 |
| 48 | Informer | LLAMA2 | **0.46** | 0.45 | 0.01 |
| 48 | PatchTST | BERT | 0.41 | **0.44** | -0.04 |
| 48 | PatchTST | GPT2 | 0.41 | **0.44** | -0.04 |
| 48 | PatchTST | LLAMA2 | 0.41 | **0.44** | -0.04 |
| 48 | Reformer | BERT | 0.44 | **0.44** | -0.01 |
| 48 | Reformer | GPT2 | 0.44 | **0.46** | -0.02 |
| 48 | Reformer | LLAMA2 | 0.42 | **0.42** | -0.0 |
| 96 | DLinear | BERT | 0.42 | **0.51** | -0.08 |
| 96 | DLinear | GPT2 | 0.42 | **0.51** | -0.09 |
| 96 | DLinear | LLAMA2 | 0.42 | **0.51** | -0.09 |
| 96 | FiLM | BERT | 0.42 | **0.49** | -0.08 |
| 96 | FiLM | GPT2 | 0.41 | **0.49** | -0.08 |
| 96 | FiLM | LLAMA2 | 0.42 | **0.49** | -0.08 |
| 96 | Informer | BERT | 0.46 | **0.47** | -0.01 |
| 96 | Informer | GPT2 | **0.49** | 0.47 | 0.02 |
| 96 | Informer | LLAMA2 | 0.47 | **0.48** | -0.01 |
| 96 | PatchTST | BERT | 0.41 | **0.47** | -0.06 |
| 96 | PatchTST | GPT2 | 0.41 | **0.47** | -0.06 |
| 96 | PatchTST | LLAMA2 | 0.41 | **0.47** | -0.05 |
| 96 | Reformer | BERT | **0.45** | 0.44 | 0.01 |
| 96 | Reformer | GPT2 | 0.45 | **0.47** | -0.02 |
| 96 | Reformer | LLAMA2 | 0.44 | **0.46** | -0.02 |
| 192 | DLinear | BERT | 0.42 | **0.57** | -0.14 |
| 192 | DLinear | GPT2 | 0.43 | **0.56** | -0.14 |
| 192 | DLinear | LLAMA2 | 0.42 | **0.56** | -0.14 |
| 192 | FiLM | BERT | 0.41 | **0.52** | -0.11 |
| 192 | FiLM | GPT2 | 0.41 | **0.52** | -0.11 |
| 192 | FiLM | LLAMA2 | 0.42 | **0.51** | -0.09 |
| 192 | Informer | BERT | **0.51** | 0.5 | 0.01 |
| 192 | Informer | GPT2 | 0.46 | **0.48** | -0.02 |
| 192 | Informer | LLAMA2 | **0.49** | 0.47 | 0.01 |
| 192 | PatchTST | BERT | 0.41 | **0.5** | -0.09 |
| 192 | PatchTST | GPT2 | 0.41 | **0.51** | -0.1 |
| 192 | PatchTST | LLAMA2 | 0.41 | **0.48** | -0.07 |
| 192 | Reformer | BERT | 0.44 | **0.47** | -0.03 |
| 192 | Reformer | GPT2 | 0.43 | **0.46** | -0.03 |
| 192 | Reformer | LLAMA2 | 0.45 | **0.46** | -0.01 |
| 336 | DLinear | BERT | 0.42 | **0.5** | -0.09 |
| 336 | DLinear | GPT2 | 0.42 | **0.5** | -0.09 |
| 336 | DLinear | LLAMA2 | 0.43 | **0.5** | -0.07 |
| 336 | FiLM | BERT | 0.42 | **0.49** | -0.07 |
| 336 | FiLM | GPT2 | 0.42 | **0.49** | -0.07 |
| 336 | FiLM | LLAMA2 | 0.42 | **0.49** | -0.07 |
| 336 | Informer | BERT | **0.48** | 0.47 | 0.02 |
| 336 | Informer | GPT2 | **0.48** | 0.47 | 0.01 |
| 336 | Informer | LLAMA2 | **0.49** | 0.47 | 0.02 |
| 336 | PatchTST | BERT | 0.41 | **0.47** | -0.05 |
| 336 | PatchTST | GPT2 | 0.41 | **0.47** | -0.06 |
| 336 | PatchTST | LLAMA2 | 0.42 | **0.47** | -0.05 |
| 336 | Reformer | BERT | 0.44 | **0.46** | -0.02 |
| 336 | Reformer | GPT2 | 0.44 | **0.46** | -0.02 |
| 336 | Reformer | LLAMA2 | 0.43 | **0.44** | -0.0 |

Table 13: Pairwise MSE comparison for the `Public Health` domain.

| Horizon | TSF-N Expert | TSF-T Expert | GMM-TS (MSE) | TimeMMD (MSE) | Delta |
|---|---|---|---|---|---|
| 12 | DLinear | GPT3.5 | **1.2781** | 1.5718 | -0.29 |
| 12 | DLinear | GPT2 | **1.2755** | 1.5728 | -0.3 |
| 12 | DLinear | LLAMA2 | **1.2542** | 1.5686 | -0.31 |
| 12 | FiLM | GPT3.5 | **1.2450** | 1.8902 | -0.65 |
| 12 | FiLM | GPT2 | **1.2444** | 1.8904 | -0.65 |
| 12 | FiLM | LLAMA2 | **1.1605** | 1.8962 | -0.74 |
| 12 | Informer | GPT3.5 | **1.0626** | 1.1597 | -0.1 |
| 12 | Informer | GPT2 | 1.0385 | **1.0100** | 0.03 |
| 12 | Informer | LLAMA2 | 1.2496 | **1.0132** | 0.24 |
| 12 | PatchTST | GPT3.5 | **0.8285** | 0.8862 | -0.06 |
| 12 | PatchTST | GPT2 | **0.8272** | 0.8858 | -0.06 |
| 12 | PatchTST | LLAMA2 | **0.8282** | 0.9198 | -0.09 |
| 12 | Reformer | GPT3.5 | **1.0117** | 1.3198 | -0.31 |
| 12 | Reformer | GPT2 | **1.0746** | 1.3032 | -0.23 |
| 12 | Reformer | LLAMA2 | 1.1247 | **1.0541** | 0.07 |
| 24 | DLinear | GPT3.5 | **1.3307** | 1.6379 | -0.31 |
| 24 | DLinear | GPT2 | **1.3350** | 1.6379 | -0.3 |
| 24 | DLinear | LLAMA2 | **1.3273** | 1.6327 | -0.31 |
| 24 | FiLM | GPT3.5 | **1.3346** | 1.7341 | -0.4 |
| 24 | FiLM | GPT2 | **1.2983** | 1.7336 | -0.44 |
| 24 | FiLM | LLAMA2 | **1.3443** | 1.7694 | -0.43 |
| 24 | Informer | GPT3.5 | **1.3144** | 1.4278 | -0.11 |
| 24 | Informer | GPT2 | **1.1808** | 1.4354 | -0.25 |
| 24 | Informer | LLAMA2 | 1.3456 | **1.2494** | 0.1 |
| 24 | PatchTST | GPT3.5 | **1.1401** | 1.4274 | -0.29 |
| 24 | PatchTST | GPT2 | **1.1373** | 1.4267 | -0.29 |
| 24 | PatchTST | LLAMA2 | **1.1287** | 1.3324 | -0.2 |
| 24 | Reformer | GPT3.5 | **1.2683** | 1.2749 | -0.01 |
| 24 | Reformer | GPT2 | **1.2664** | 1.2732 | -0.01 |
| 24 | Reformer | LLAMA2 | **1.1444** | 1.3164 | -0.17 |
| 36 | DLinear | GPT3.5 | **1.3606** | 1.6339 | -0.27 |
| 36 | DLinear | GPT2 | **1.3916** | 1.6337 | -0.24 |
| 36 | DLinear | LLAMA2 | **1.3667** | 1.6314 | -0.26 |
| 36 | FiLM | GPT3.5 | **1.3396** | 1.6919 | -0.35 |
| 36 | FiLM | GPT2 | **1.3396** | 1.6918 | -0.35 |
| 36 | FiLM | LLAMA2 | **1.3486** | 1.6797 | -0.33 |
| 36 | Informer | GPT3.5 | **1.2517** | 1.4964 | -0.24 |
| 36 | Informer | GPT3.5 | **1.2517** | 1.5301 | -0.28 |
| 36 | Informer | GPT2 | **1.3061** | 1.5267 | -0.22 |
| 36 | Informer | LLAMA2 | **1.3360** | 1.4404 | -0.1 |
| 36 | PatchTST | GPT3.5 | **1.3303** | 1.6323 | -0.3 |
| 36 | PatchTST | GPT2 | **1.3105** | 1.6329 | -0.32 |
| 36 | PatchTST | LLAMA2 | **1.3181** | 1.6132 | -0.3 |
| 36 | Reformer | GPT3.5 | **1.2727** | 1.3266 | -0.05 |
| 36 | Reformer | GPT2 | **1.2604** | 1.3325 | -0.07 |
| 36 | Reformer | LLAMA2 | **1.2960** | 1.4491 | -0.15 |
| 48 | DLinear | GPT3.5 | **1.4695** | 1.7188 | -0.25 |
| 48 | DLinear | GPT2 | **1.4659** | 1.7188 | -0.25 |
| 48 | DLinear | LLAMA2 | **1.4410** | 1.6834 | -0.24 |
| 48 | FiLM | GPT3.5 | **1.3941** | 1.7494 | -0.36 |
| 48 | FiLM | GPT2 | **1.3882** | 1.7467 | -0.36 |
| 48 | FiLM | LLAMA2 | **1.4095** | 1.7578 | -0.35 |
| 48 | Informer | GPT3.5 | **1.4794** | 1.6853 | -0.21 |
| 48 | Informer | GPT2 | **1.3917** | 1.6759 | -0.28 |
| 48 | Informer | LLAMA2 | **1.3763** | 1.6073 | -0.23 |
| 48 | PatchTST | GPT3.5 | **1.3858** | 1.8986 | -0.51 |
| 48 | PatchTST | GPT2 | **1.4246** | 1.8978 | -0.47 |
| 48 | PatchTST | LLAMA2 | **1.3948** | 1.7737 | -0.38 |
| 48 | Reformer | GPT3.5 | **1.3376** | 1.4628 | -0.13 |
| 48 | Reformer | GPT2 | **1.4128** | 1.4535 | -0.04 |
| 48 | Reformer | LLAMA2 | **1.3712** | 1.4464 | -0.08 |

Table 14: Pairwise MSE comparison for the `Security` domain.

| Horizon | TSF-N Expert | TSF-T Expert | GMM-TS (MSE) | TimeMMD (MSE) | Delta |
|---|---|---|---|---|---|
| 6 | DLinear | GPT3.5 | **102.9184** | 103.2177 | -0.3 |
| 6 | DLinear | GPT2 | **102.9638** | 103.2089 | -0.25 |
| 6 | DLinear | LLAMA2 | **102.9615** | 103.1953 | -0.23 |
| 6 | FiLM | GPT3.5 | **110.6303** | 114.7459 | -4.12 |
| 6 | FiLM | GPT2 | **111.4065** | 114.7540 | -3.35 |
| 6 | FiLM | LLAMA2 | **109.1073** | 114.8041 | -5.7 |
| 6 | Informer | GPT3.5 | 126.4486 | **124.9059** | 1.54 |
| 6 | Informer | GPT2 | 126.1993 | **124.8889** | 1.31 |
| 6 | Informer | LLAMA2 | **125.3303** | 126.0265 | -0.7 |
| 6 | PatchTST | GPT3.5 | **106.6492** | 109.1538 | -2.5 |
| 6 | PatchTST | GPT2 | **106.7367** | 109.1556 | -2.42 |
| 6 | PatchTST | LLAMA2 | **105.8426** | 108.0277 | -2.19 |
| 6 | Reformer | GPT3.5 | **122.7506** | 127.8057 | -5.06 |
| 6 | Reformer | GPT2 | **121.9643** | 122.6299 | -0.67 |
| 6 | Reformer | LLAMA2 | 125.3047 | **119.4994** | 5.81 |
| 8 | DLinear | GPT3.5 | **105.6411** | 107.7553 | -2.11 |
| 8 | DLinear | GPT2 | **105.7126** | 107.7545 | -2.04 |
| 8 | DLinear | LLAMA2 | **105.4353** | 107.7964 | -2.36 |
| 8 | FiLM | GPT3.5 | **108.8374** | 109.4419 | -0.6 |
| 8 | FiLM | GPT2 | **108.7304** | 109.4354 | -0.7 |
| 8 | FiLM | LLAMA2 | **107.8102** | 109.0228 | -1.21 |
| 8 | Informer | GPT3.5 | **126.7646** | 127.2815 | -0.52 |
| 8 | Informer | GPT2 | **126.9326** | 127.3436 | -0.41 |
| 8 | Informer | LLAMA2 | **126.4637** | 127.0559 | -0.59 |
| 8 | PatchTST | GPT3.5 | 114.0695 | **111.6167** | 2.45 |
| 8 | PatchTST | GPT2 | 114.7777 | **112.1871** | 2.59 |
| 8 | PatchTST | LLAMA2 | **109.8454** | 110.8373 | -0.99 |
| 8 | Reformer | GPT3.5 | **124.6640** | 127.3454 | -2.68 |
| 8 | Reformer | GPT2 | **123.2080** | 127.3009 | -4.09 |
| 8 | Reformer | LLAMA2 | **121.0255** | 121.4871 | -0.46 |
| 10 | DLinear | GPT3.5 | **107.3247** | 109.7904 | -2.47 |
| 10 | DLinear | GPT2 | **107.2617** | 109.7907 | -2.53 |
| 10 | DLinear | LLAMA2 | **108.1359** | 109.7591 | -1.62 |
| 10 | FiLM | GPT3.5 | **110.3583** | 110.9249 | -0.57 |
| 10 | FiLM | GPT2 | **110.0805** | 110.9051 | -0.82 |
| 10 | FiLM | LLAMA2 | **108.3393** | 111.3900 | -3.05 |
| 10 | Informer | GPT3.5 | 131.2260 | **126.6200** | 4.61 |
| 10 | Informer | GPT2 | 130.8043 | **126.6540** | 4.15 |
| 10 | Informer | LLAMA2 | **128.0662** | 128.8640 | -0.8 |
| 10 | PatchTST | GPT3.5 | **109.9814** | 116.0471 | -6.07 |
| 10 | PatchTST | GPT2 | **110.0050** | 114.9653 | -4.96 |
| 10 | PatchTST | LLAMA2 | 113.6006 | **113.0154** | 0.59 |
| 10 | Reformer | GPT3.5 | **121.1815** | 127.1552 | -5.97 |
| 10 | Reformer | GPT2 | **121.0420** | 127.0808 | -6.04 |
| 10 | Reformer | LLAMA2 | **116.2173** | 122.5238 | -6.31 |
| 12 | DLinear | GPT3.5 | **108.6712** | 111.2703 | -2.6 |
| 12 | DLinear | GPT2 | **108.7270** | 111.2681 | -2.54 |
| 12 | DLinear | LLAMA2 | **108.9626** | 111.2568 | -2.29 |
| 12 | FiLM | GPT3.5 | **111.0610** | 113.1494 | -2.09 |
| 12 | FiLM | GPT2 | **109.7006** | 113.0860 | -3.39 |
| 12 | FiLM | LLAMA2 | **109.2312** | 112.4943 | -3.26 |
| 12 | Informer | GPT3.5 | 130.5862 | **128.3961** | 2.19 |
| 12 | Informer | GPT2 | 130.5437 | **128.4168** | 2.13 |
| 12 | Informer | LLAMA2 | **127.3390** | 130.8707 | -3.53 |
| 12 | PatchTST | GPT3.5 | **110.4975** | 114.7932 | -4.3 |
| 12 | PatchTST | GPT2 | **110.4948** | 113.6508 | -3.16 |
| 12 | PatchTST | LLAMA2 | **110.2060** | 113.9373 | -3.73 |
| 12 | Reformer | GPT3.5 | **121.3439** | 126.9580 | -5.61 |
| 12 | Reformer | GPT2 | **121.0127** | 126.8623 | -5.85 |
| 12 | Reformer | LLAMA2 | 120.6723 | **117.7383** | 2.93 |

Table 15: Pairwise MSE comparison for the `SocialGood` domain.

| Horizon | TSF-N Expert | TSF-T Expert | GMM-TS (MSE) | TimeMMD (MSE) | Delta |
|---|---|---|---|---|---|
| 6 | DLinear | GPT3.5 | **0.9188** | 0.9682 | -0.05 |
| 6 | DLinear | GPT2 | **0.9189** | 0.9658 | -0.05 |
| 6 | DLinear | LLAMA2 | **0.9611** | 0.9703 | -0.01 |
| 6 | FiLM | GPT3.5 | **0.9318** | 0.9556 | -0.02 |
| 6 | FiLM | GPT2 | **0.9291** | 0.9555 | -0.03 |
| 6 | FiLM | LLAMA2 | **0.9412** | 0.9486 | -0.01 |
| 6 | Informer | GPT3.5 | **0.7583** | 0.8545 | -0.1 |
| 6 | Informer | GPT2 | **0.7311** | 0.8285 | -0.1 |
| 6 | Informer | LLAMA2 | 0.8140 | **0.7681** | 0.05 |
| 6 | PatchTST | GPT3.5 | 0.8587 | **0.8287** | 0.03 |
| 6 | PatchTST | GPT2 | **0.8798** | 1.1037 | -0.22 |
| 6 | PatchTST | LLAMA2 | 0.8735 | **0.8339** | 0.04 |
| 6 | Reformer | GPT3.5 | 0.7826 | **0.7794** | 0 |
| 6 | Reformer | GPT2 | **0.8059** | 0.8615 | -0.06 |
| 6 | Reformer | LLAMA2 | **0.8043** | 0.8382 | -0.03 |
| 8 | DLinear | GPT3.5 | 0.9686 | **0.9388** | 0.03 |
| 8 | DLinear | GPT2 | 0.9647 | **0.9395** | 0.03 |
| 8 | DLinear | LLAMA2 | **0.9601** | 0.9757 | -0.02 |
| 8 | FiLM | GPT3.5 | 1.0110 | **1.0093** | 0 |
| 8 | FiLM | GPT2 | 1.0132 | **0.9938** | 0.02 |
| 8 | FiLM | LLAMA2 | **1.0027** | 1.0300 | -0.03 |
| 8 | Informer | GPT3.5 | 0.8292 | **0.7453** | 0.08 |
| 8 | Informer | GPT2 | 0.7627 | **0.7566** | 0.01 |
| 8 | Informer | LLAMA2 | **0.7555** | 0.8990 | -0.14 |
| 8 | PatchTST | GPT3.5 | **0.8886** | 1.0846 | -0.2 |
| 8 | PatchTST | GPT2 | **0.8931** | 1.0081 | -0.11 |
| 8 | PatchTST | LLAMA2 | **0.9775** | 1.0432 | -0.07 |
| 8 | Reformer | GPT3.5 | **0.9104** | 0.9535 | -0.04 |
| 8 | Reformer | GPT2 | 0.8959 | **0.8554** | 0.04 |
| 8 | Reformer | LLAMA2 | **0.8754** | 0.9465 | -0.07 |
| 10 | DLinear | GPT3.5 | 1.0571 | **1.0147** | 0.04 |
| 10 | DLinear | GPT2 | 1.0505 | **1.0148** | 0.04 |
| 10 | DLinear | LLAMA2 | 1.0886 | **1.0370** | 0.05 |
| 10 | FiLM | GPT3.5 | **1.0748** | 1.1038 | -0.03 |
| 10 | FiLM | GPT2 | **1.0806** | 1.0957 | -0.02 |
| 10 | FiLM | LLAMA2 | 1.0931 | **1.0828** | 0.01 |
| 10 | Informer | GPT3.5 | 0.9705 | **0.9118** | 0.06 |
| 10 | Informer | GPT2 | 0.9005 | **0.8786** | 0.02 |
| 10 | Informer | LLAMA2 | **0.8110** | 0.8508 | -0.04 |
| 10 | PatchTST | GPT3.5 | 0.9914 | **0.9903** | 0 |
| 10 | PatchTST | GPT2 | **0.9828** | 1.0183 | -0.04 |
| 10 | PatchTST | LLAMA2 | 1.0614 | **0.9683** | 0.09 |
| 10 | Reformer | GPT3.5 | 0.9864 | **0.9460** | 0.04 |
| 10 | Reformer | GPT2 | **0.9452** | 1.0200 | -0.07 |
| 10 | Reformer | LLAMA2 | 1.0611 | **0.9825** | 0.08 |
| 12 | DLinear | GPT3.5 | **1.1033** | 1.1517 | -0.05 |
| 12 | DLinear | GPT2 | **1.1008** | 1.1519 | -0.05 |
| 12 | DLinear | LLAMA2 | **1.0989** | 1.1468 | -0.05 |
| 12 | FiLM | GPT3.5 | **1.1181** | 1.1695 | -0.05 |
| 12 | FiLM | GPT2 | **1.1124** | 1.1652 | -0.05 |
| 12 | FiLM | LLAMA2 | **1.1505** | 1.1594 | -0.01 |
| 12 | Informer | GPT3.5 | **0.8410** | 0.9560 | -0.12 |
| 12 | Informer | GPT2 | **0.8402** | 0.9557 | -0.12 |
| 12 | Informer | LLAMA2 | **0.9147** | 0.9635 | -0.05 |
| 12 | PatchTST | GPT3.5 | 1.0621 | **1.0419** | 0.02 |
| 12 | PatchTST | GPT2 | **1.0478** | 1.0714 | -0.02 |
| 12 | PatchTST | LLAMA2 | **1.0463** | 1.1580 | -0.11 |
| 12 | Reformer | GPT3.5 | 1.1757 | **1.1040** | 0.07 |
| 12 | Reformer | GPT2 | 1.1832 | **1.0935** | 0.09 |
| 12 | Reformer | LLAMA2 | 1.0301 | **0.9680** | 0.06 |

Table 16: Pairwise MSE comparison for the `Traffic` domain.

| Horizon | TSF-N Expert | TSF-T Expert | GMM-TS (MSE) | TimeMMD (MSE) | Delta |
|---|---|---|---|---|---|
| 6 | DLinear | GPT3.5 | **0.2109** | 0.2434 | -0.03 |
| 6 | DLinear | GPT2 | **0.2096** | 0.2434 | -0.03 |
| 6 | DLinear | LLAMA2 | **0.1948** | 0.2398 | -0.04 |
| 6 | FiLM | GPT3.5 | **0.1926** | 0.2259 | -0.03 |
| 6 | FiLM | GPT2 | **0.1871** | 0.2259 | -0.04 |
| 6 | FiLM | LLAMA2 | **0.1948** | 0.2251 | -0.03 |
| 6 | Informer | GPT3.5 | **0.1569** | 0.1765 | -0.02 |
| 6 | Informer | GPT2 | **0.1562** | 0.1800 | -0.02 |
| 6 | Informer | LLAMA2 | **0.1600** | 0.1947 | -0.03 |
| 6 | PatchTST | GPT3.5 | **0.1638** | 0.1781 | -0.01 |
| 6 | PatchTST | GPT2 | **0.1709** | 0.1746 | -0 |
| 6 | PatchTST | LLAMA2 | **0.1662** | 0.1740 | -0.01 |
| 6 | Reformer | GPT3.5 | **0.1941** | 0.1981 | -0 |
| 6 | Reformer | GPT2 | **0.1814** | 0.2010 | -0.02 |
| 6 | Reformer | LLAMA2 | **0.1775** | 0.2263 | -0.05 |
| 8 | DLinear | GPT3.5 | **0.1953** | 0.2871 | -0.09 |
| 8 | DLinear | GPT2 | **0.1785** | 0.2880 | -0.11 |
| 8 | DLinear | LLAMA2 | **0.1790** | 0.2871 | -0.11 |
| 8 | FiLM | GPT3.5 | **0.1845** | 0.2249 | -0.04 |
| 8 | FiLM | GPT2 | **0.1844** | 0.2249 | -0.04 |
| 8 | FiLM | LLAMA2 | **0.1836** | 0.2249 | -0.04 |
| 8 | Informer | GPT3.5 | 0.1765 | **0.1761** | 0 |
| 8 | Informer | GPT2 | 0.1788 | **0.1752** | 0 |
| 8 | Informer | LLAMA2 | **0.1588** | 0.1847 | -0.03 |
| 8 | PatchTST | GPT3.5 | **0.1758** | 0.1786 | -0 |
| 8 | PatchTST | GPT2 | **0.1738** | 0.1791 | -0.01 |
| 8 | PatchTST | LLAMA2 | **0.1856** | 0.1863 | -0 |
| 8 | Reformer | GPT3.5 | **0.1952** | 0.2007 | -0.01 |
| 8 | Reformer | GPT2 | **0.1937** | 0.1972 | -0 |
| 8 | Reformer | LLAMA2 | **0.1815** | 0.2068 | -0.03 |
| 10 | DLinear | GPT3.5 | **0.2019** | 0.2360 | -0.03 |
| 10 | DLinear | GPT2 | **0.2014** | 0.2359 | -0.03 |
| 10 | DLinear | LLAMA2 | **0.1901** | 0.2351 | -0.04 |
| 10 | FiLM | GPT3.5 | **0.1749** | 0.2225 | -0.05 |
| 10 | FiLM | GPT2 | **0.1754** | 0.2224 | -0.05 |
| 10 | FiLM | LLAMA2 | **0.1732** | 0.2232 | -0.05 |
| 10 | Informer | GPT3.5 | **0.1608** | 0.1831 | -0.02 |
| 10 | Informer | GPT2 | **0.1691** | 0.1848 | -0.02 |
| 10 | Informer | LLAMA2 | **0.1690** | 0.1950 | -0.03 |
| 10 | PatchTST | GPT3.5 | **0.1829** | 0.1904 | -0.01 |
| 10 | PatchTST | GPT2 | **0.1818** | 0.1902 | -0.01 |
| 10 | PatchTST | LLAMA2 | **0.1727** | 0.1946 | -0.02 |
| 10 | Reformer | GPT3.5 | **0.1808** | 0.2241 | -0.04 |
| 10 | Reformer | GPT2 | **0.1842** | 0.2168 | -0.03 |
| 10 | Reformer | LLAMA2 | **0.1784** | 0.2391 | -0.06 |
| 12 | DLinear | GPT3.5 | **0.2274** | 0.2591 | -0.03 |
| 12 | DLinear | GPT2 | **0.2269** | 0.2590 | -0.03 |
| 12 | DLinear | LLAMA2 | **0.2396** | 0.2611 | -0.02 |
| 12 | FiLM | GPT3.5 | **0.2271** | 0.2685 | -0.04 |
| 12 | FiLM | GPT2 | **0.2271** | 0.2684 | -0.04 |
| 12 | FiLM | LLAMA2 | **0.2379** | 0.2685 | -0.03 |
| 12 | Informer | GPT3.5 | **0.2049** | 0.2277 | -0.02 |
| 12 | Informer | GPT2 | **0.2036** | 0.2289 | -0.03 |
| 12 | Informer | LLAMA2 | **0.1978** | 0.2069 | -0.01 |
| 12 | PatchTST | GPT3.5 | **0.2421** | 0.2575 | -0.02 |
| 12 | PatchTST | GPT2 | **0.2387** | 0.2573 | -0.02 |
| 12 | PatchTST | LLAMA2 | **0.2326** | 0.2556 | -0.02 |
| 12 | Reformer | GPT3.5 | **0.2225** | 0.2340 | -0.01 |
| 12 | Reformer | GPT2 | **0.2210** | 0.2281 | -0.01 |
| 12 | Reformer | LLAMA2 | **0.2160** | 0.2703 | -0.05 |

Table 17: Unimodal forecasting results for the `Energy` domain. Detailed per-horizon, per-expert results complement the aggregated domain-level averages reported in Table 1.

| Horizon | Expert | Expert Type | MAE | MSE | RMSE | MAPE | MSPE |
|---|---|---|---|---|---|---|---|
| 12 | Informer | TSF-N | 0.283 | 0.140 | 0.374 | 1.529 | 33.078 |
| 12 | Reformer | TSF-N | 0.382 | 0.263 | 0.513 | 1.418 | 22.324 |
| 12 | DLinear | TSF-N | 0.359 | 0.221 | 0.471 | 1.323 | 15.036 |
| 12 | PatchTST | TSF-N | 0.227 | 0.110 | 0.332 | 0.991 | 20.750 |
| 12 | FiLM | TSF-N | 0.351 | 0.207 | 0.455 | 1.417 | 18.193 |
| 12 | GPT3.5 | TSF-T | 1.047 | 1.642 | 1.281 | 5.901 | 674.482 |
| 24 | Informer | TSF-N | 0.412 | 0.278 | 0.528 | 2.140 | 107.619 |
| 24 | Reformer | TSF-N | 0.537 | 0.470 | 0.685 | 2.235 | 98.262 |
| 24 | DLinear | TSF-N | 0.484 | 0.416 | 0.645 | 1.869 | 50.715 |
| 24 | PatchTST | TSF-N | 0.386 | 0.276 | 0.525 | 1.361 | 41.275 |
| 24 | FiLM | TSF-N | 0.474 | 0.408 | 0.639 | 1.774 | 46.942 |
| 24 | GPT3.5 | TSF-T | 1.330 | 2.463 | 1.570 | 7.693 | 1372.181 |

Table 18: Unimodal forecasting results for the `Public Health` domain. Detailed per-horizon, per-expert results complement the aggregated domain-level averages reported in Table 1.

| Horizon | Expert | Expert Type | MAE | MSE | RMSE | MAPE | MSPE |
|---|---|---|---|---|---|---|---|
| 12 | Informer | TSF-N | 0.623 | 0.499 | 0.706 | 1.225 | 4.928 |
| 12 | Reformer | TSF-N | 0.708 | 0.632 | 0.795 | 1.372 | 6.276 |
| 12 | DLinear | TSF-N | 0.650 | 0.541 | 0.735 | 1.304 | 5.168 |
| 12 | PatchTST | TSF-N | 0.548 | 0.408 | 0.639 | 1.088 | 3.948 |
| 12 | FiLM | TSF-N | 0.681 | 0.581 | 0.762 | 1.323 | 5.506 |
| 12 | GPT3.5 | TSF-T | 1.304 | 1.983 | 1.408 | 3.987 | 78.280 |
| 24 | Informer | TSF-N | 0.766 | 0.676 | 0.822 | 1.645 | 14.449 |
| 24 | Reformer | TSF-N | 0.794 | 0.705 | 0.840 | 1.603 | 13.348 |
| 24 | DLinear | TSF-N | 0.732 | 0.648 | 0.805 | 1.593 | 11.824 |
| 24 | PatchTST | TSF-N | 0.649 | 0.558 | 0.747 | 1.341 | 9.807 |
| 24 | FiLM | TSF-N | 0.756 | 0.671 | 0.819 | 1.484 | 10.808 |
| 24 | GPT3.5 | TSF-T | 1.528 | 2.607 | 1.614 | 4.980 | 152.014 |

Table 19: The MSE achieved when training our method with expert pairs, triplets and quartets. For each domain, we report the average across pairs (Pairs MSE), triplets (Triplets MSE) and quartets of experts.

| Domain | Pairs MSE | Triplets MSE | Quartets MSE |
|---|---|---|---|
| Climate | 1.00 | 1.02 | 1.03 |
| Energy | 0.34 | 0.30 | 0.27 |
| Public | 1.27 | 1.21 | 1.24 |
| Traffic | 0.19 | 0.18 | 0.18 |

Table 20: Using triplets of experts to approximate best performing expert pairs. Given two TSF-N experts: $e_n^1$ and $e_n^2$ and a single TSF-T expert $e_t$, we evaluate the performance of the pairs: $(e_n^1, e_t)$ and $(e_n^2, e_t)$ and of the triplet $(e_n^1, e_n^2, e_t)$. We report the average MSE of the two pairs and compare it to the MSE of the respective triplet. The MSE of triplets of experts fused with our gating architecture, consistently surpasses the pair MSE average.

| Domain | $e_t$ | $e_n^1$ | $e_n^2$ | Avg. Pair MSE | Avg. Triplet MSE |
|---|---|---|---|---|---|
| Agriculture | GPT3.5 | DLinear | Informer | 0.24 | 0.22 |
| | | DLinear | PatchTST | 0.10 | 0.09 |
| | | Informer | PatchTST | 0.23 | 0.15 |
| Climate | GPT3.5 | DLinear | Informer | 0.99 | 1.03 |
| | | DLinear | PatchTST | 1.00 | 1.00 |
| | | Informer | PatchTST | 1.01 | 1.01 |
| Economy | GPT3.5 | DLinear | Informer | 0.30 | 0.13 |
| | | DLinear | PatchTST | 0.03 | 0.02 |
| | | Informer | PatchTST | 0.29 | 0.11 |
| Energy | GPT3.5 | DLinear | Informer | 0.38 | 0.36 |
| | | DLinear | PatchTST | 0.31 | 0.27 |
| | | Informer | PatchTST | 0.34 | 0.27 |
| Public Health | GPT3.5 | DLinear | Informer | 1.32 | 1.29 |
| | | DLinear | PatchTST | 1.27 | 1.15 |
| | | Informer | PatchTST | 1.22 | 1.19 |
| Security | GPT3.5 | DLinear | Informer | 117.45 | 113.45 |
| | | DLinear | PatchTST | 108.22 | 108.43 |
| | | Informer | PatchTST | 119.53 | 116.54 |
| Social Good | GPT3.5 | DLinear | Informer | 0.93 | 0.83 |
| | | DLinear | PatchTST | 0.98 | 0.98 |
| | | Informer | PatchTST | 0.90 | 0.89 |
| Traffic | GPT3.5 | DLinear | Informer | 0.19 | 0.17 |
| | | DLinear | PatchTST | 0.20 | 0.19 |
| | | Informer | PatchTST | 0.18 | 0.18 |

Table 21: Comparison of TimeMMD and GMM-TS in the offline pre-training and joint training regimes. We report the average MSE for each domain, across experts combinations and horizon lengths. For each domain, the best MSE is highlighted in bold. The second-best MSE is underlined.

| Domain | TimeMMD | GMM-TS with Offline Pretraining | GMM-TS with Joint Training |
|---|---|---|---|
| Agriculture | 0.11 | 0.10 | **0.09** |
| Climate | 1.15 | 1.19 | **1.02** |
| Economy | 0.04 | **0.02** | **0.02** |
| Energy | 0.29 | 0.28 | **0.27** |
| Environment | 0.47 | 0.48 | **0.41** |
| Public Health | 1.46 | 1.56 | **1.17** |
| Security | 112.28 | 112.28 | **110.30** |
| Social Good | 1.09 | 1.01 | **0.95** |
| Traffic | 0.21 | 0.20 | **0.19** |

Table 22: MSE of Different TSF-N and TSF-T multi-expert combinations across domains, when using the direct, latent and hierarchical aggregation methods. Abbreviations: D (DLinear), I (Informer), P (PatchTST), R (Reformer). For each combination (row), we highlight the best performing aggregation in bold.

| Domain | TSF-T | TSF-N | | | | Direct Agg. | Hierarchical Agg. | Latent Agg. |
|---|---|---|---|---|---|---|---|---|
| | | D | I | R | P | | | |
| Economy | GPT3.5 | + | + | | + | **0.22** | 0.26 | 1.29 |
| | | | + | + | + | **0.16** | 0.21 | 1.40 |
| | | + | + | + | + | **0.12** | 0.16 | 1.00 |
| Energy | GPT3.5 | + | + | | + | 0.35 | 0.36 | **0.31** |
| | | | + | + | + | **0.29** | 0.30 | 0.32 |
| | | + | + | + | + | 0.29 | 0.29 | **0.28** |
| Public Health | GPT3.5 | + | + | | + | 1.30 | **1.27** | 1.31 |
| | | | + | + | + | 1.19 | **1.18** | 1.28 |
| | | + | + | + | + | 1.22 | **1.17** | 1.28 |
| Security | GPT3.5 | + | + | | + | **114.04** | 116.96 | 127.78 |
| | | | + | + | + | **117.32** | 119.01 | 128.10 |
| | | + | + | + | + | 115.50 | **114.07** | 127.47 |
| SocialGood | GPT3.5 | + | + | | + | 0.86 | **0.84** | 0.86 |
| | | | + | + | + | 0.85 | 0.86 | **0.82** |
| | | + | + | + | + | **0.85** | 0.90 | **0.85** |
| Traffic | GPT3.5 | + | + | | + | **0.18** | **0.18** | **0.18** |
| | | | + | + | + | 0.18 | **0.17** | 0.18 |
| | | + | + | + | + | **0.17** | **0.17** | 0.18 |

Table 23: MSE of Different TSF-N and TSF-T experts pairs across domains, when using the direct and latent aggregation methods.

| Domain | TSF-T | TSF-N | Direct Agg. | Latent Agg. |
|---|---|---|---|---|
| Economy | GPT3.5 | DLinear | 0.03 | 0.77 |
| | | FiLM | 0.02 | 8.26 |
| | | Informer | 0.56 | 1.03 |
| | | PatchTST | 0.02 | 8.02 |
| | | Reformer | 0.28 | 1.43 |
| Energy | GPT3.5 | DLinear | 0.35 | 0.30 |
| | | FiLM | 0.35 | 1.05 |
| | | Informer | 0.41 | 0.30 |
| | | PatchTST | 0.27 | 1.10 |
| | | Reformer | 0.43 | 0.27 |
| Public Health | GPT3.5 | DLinear | 1.36 | 1.58 |
| | | FiLM | 1.33 | 1.54 |
| | | Informer | 1.28 | 1.34 |
| | | PatchTST | 1.17 | 1.52 |
| | | Reformer | 1.22 | 1.23 |
| Security | GPT3.5 | DLinear | 106.14 | 123.64 |
| | | FiLM | 110.22 | 133.50 |
| | | Informer | 128.76 | 128.96 |
| | | PatchTST | 110.30 | 132.81 |
| | | Reformer | 122.48 | 128.40 |
| SocialGood | GPT3.5 | DLinear | 1.01 | 0.90 |
| | | FiLM | 1.03 | 1.85 |
| | | Informer | 0.85 | 0.84 |
| | | PatchTST | 0.95 | 1.84 |
| | | Reformer | 0.96 | 0.90 |
| Traffic | GPT3.5 | DLinear | 0.21 | 0.27 |
| | | FiLM | 0.19 | 1.22 |
| | | Informer | 0.17 | 0.19 |
| | | PatchTST | 0.19 | 1.06 |
| | | Reformer | 0.20 | 0.20 |

Table 24: MSE of Different TSF-N and TSF-T experts pairs for representative domains, when varying on the gating dimension.

| Domain | TSF-T | TSF-N | Gating Dim. | | |
|---|---|---|---|---|---|
| | | | 128 | 256 | 512 |
| Economy | GPT3.5 | DLinear | 0.03 | 0.03 | 0.03 |
| | | FiLM | 0.03 | 0.02 | 0.02 |
| | | Informer | 0.55 | 0.56 | 0.61 |
| | | PatchTST | 0.02 | 0.02 | 0.02 |
| | | Reformer | 0.40 | 0.28 | 0.33 |
| Energy | GPT3.5 | DLinear | 0.35 | 0.35 | 0.35 |
| | | FiLM | 0.35 | 0.35 | 0.35 |
| | | Informer | 0.36 | 0.41 | 0.37 |
| | | PatchTST | 0.27 | 0.27 | 0.28 |
| | | Reformer | 0.46 | 0.43 | 0.45 |
| Public Health | GPT3.5 | DLinear | 1.35 | 1.36 | 1.37 |
| | | FiLM | 1.34 | 1.33 | 1.31 |
| | | Informer | 1.22 | 1.28 | 1.27 |
| | | PatchTST | 1.17 | 1.17 | 1.17 |
| | | Reformer | 1.26 | 1.22 | 1.22 |
| Security | GPT3.5 | DLinear | 106.36 | 106.14 | 106.67 |
| | | FiLM | 108.89 | 110.22 | 109.35 |
| | | Informer | 126.75 | 128.76 | 128.18 |
| | | PatchTST | 110.40 | 110.30 | 112.06 |
| | | Reformer | 120.61 | 122.48 | 119.97 |
| SocialGood | GPT3.5 | DLinear | 1.01 | 1.01 | 1.02 |
| | | FiLM | 1.02 | 1.03 | 1.03 |
| | | Informer | 0.82 | 0.85 | 0.84 |
| | | PatchTST | 1.02 | 0.95 | 0.98 |
| | | Reformer | 0.93 | 0.96 | 0.92 |
| Traffic | GPT3.5 | DLinear | 0.20 | 0.21 | 0.20 |
| | | FiLM | 0.20 | 0.19 | 0.20 |
| | | Informer | 0.17 | 0.17 | 0.18 |
| | | PatchTST | 0.19 | 0.19 | 0.20 |
| | | Reformer | 0.20 | 0.20 | 0.20 |

| TSF-N Expert | Adaptive Gating (ours) MSE | Fixed Weight Matrix MSE | Increase (%) |
|---|---|---|---|
| DLinear | **10.4477** | 16.2892 | 55.9 |
| Informer | **21.9498** | 37.0470 | 68.7 |
| PatchTST | **21.3438** | 35.3823 | 65.7 |
| Reformer | **20.8385** | 35.0916 | 68.4 |

Table 25: Average performance degradation when removing dynamic gating, grouped by TSF-N expert. Values are averaged across all domains, horizons, and gating dimensions.

| Domain | Adaptive Gating (ours) MSE | Fixed Weight Matrix MSE | Increase (%) |
|---|---|---|---|
| Economy | **0.0255** | 0.3324 | 1272.8 |
| Energy | **0.3067** | 0.5885 | 114.6 |
| Public Health | **1.0539** | 1.2248 | 16.2 |
| Security | **106.9026** | 116.0750 | 8.6 |
| SocialGood | **0.8839** | 1.1521 | 31.1 |
| Traffic | **0.1724** | 0.3292 | 92.7 |

Table 26: Average performance degradation when removing dynamic gating, grouped by domain. Values are averaged across all TSF-N experts, horizons, and gating dimensions.

| Pred Len | Gating Dim | TSF-N Expert | TSF-T Expert | Regular MSE | Ablation MSE | Increase (%) |
|---|---|---|---|---|---|---|
| 12 | 32 | DLinear | GPT-3.5 | **0.1661** | 0.5063 | 204.9 |
| 12 | 32 | Informer | GPT-3.5 | **0.1661** | 0.4866 | 193.0 |
| 12 | 32 | Reformer | GPT-3.5 | **0.1661** | 0.4770 | 187.3 |
| 12 | 64 | DLinear | GPT-3.5 | **0.1418** | 0.5024 | 254.3 |
| 12 | 64 | Informer | GPT-3.5 | **0.1418** | 0.4627 | 226.3 |
| 12 | 64 | Reformer | GPT-3.5 | **0.1418** | 0.4813 | 239.4 |
| 24 | 32 | DLinear | GPT-3.5 | **0.2816** | 0.5808 | 106.2 |
| 24 | 32 | Informer | GPT-3.5 | **0.2816** | 0.5606 | 99.1 |
| 24 | 32 | Reformer | GPT-3.5 | **0.2816** | 0.5343 | 89.7 |
| 24 | 64 | DLinear | GPT-3.5 | **0.2435** | 0.5811 | 138.7 |
| 24 | 64 | Informer | GPT-3.5 | **0.2435** | 0.5887 | 141.8 |
| 24 | 64 | Reformer | GPT-3.5 | **0.2435** | 0.5260 | 116.0 |
| 36 | 32 | DLinear | GPT-3.5 | **0.3768** | 0.6420 | 70.4 |
| 36 | 32 | Informer | GPT-3.5 | **0.3768** | 0.5576 | 48.0 |
| 36 | 32 | Reformer | GPT-3.5 | **0.3768** | 0.6387 | 69.5 |
| 36 | 64 | DLinear | GPT-3.5 | **0.3367** | 0.6427 | 90.9 |
| 36 | 64 | Informer | GPT-3.5 | **0.3367** | 0.5836 | 73.3 |
| 36 | 64 | Reformer | GPT-3.5 | **0.3367** | 0.6551 | 94.5 |
| 48 | 32 | DLinear | GPT-3.5 | **0.4642** | 0.7467 | 60.8 |
| 48 | 32 | Informer | GPT-3.5 | **0.4642** | 0.6251 | 34.7 |
| 48 | 32 | Reformer | GPT-3.5 | **0.4642** | 0.7196 | 55.0 |
| 48 | 64 | DLinear | GPT-3.5 | **0.4429** | 0.7277 | 64.3 |
| 48 | 64 | Informer | GPT-3.5 | **0.4429** | 0.5887 | 32.9 |
| 48 | 64 | Reformer | GPT-3.5 | **0.4429** | 0.7077 | 59.8 |

Table 27: Comparison of Regular (Dynamic Gating) vs. Ablation (Fixed Weights) performance for the Energy domain. Increase (%) is relative to the lower MSE in each row.

| Pred Len | Gating Dim | TSF-N Expert | TSF-T Expert | Regular MSE | Ablation MSE | Increase (%) |
|---|---|---|---|---|---|---|
| 6 | 32 | DLinear | GPT-3.5 | **102.9655** | 103.8636 | 0.9 |
| 6 | 32 | Informer | GPT-3.5 | **102.9655** | 121.5132 | 18.0 |
| 6 | 32 | PatchTST | GPT-3.5 | **102.9655** | 111.9161 | 8.7 |
| 6 | 32 | Reformer | GPT-3.5 | **102.9655** | 118.0281 | 14.6 |
| 6 | 64 | DLinear | GPT-3.5 | **102.9404** | 103.9616 | 1.0 |
| 6 | 64 | Informer | GPT-3.5 | **102.9404** | 124.6346 | 21.1 |
| 6 | 64 | PatchTST | GPT-3.5 | **102.9404** | 109.7373 | 6.6 |
| 6 | 64 | Reformer | GPT-3.5 | **102.9404** | 122.8363 | 19.3 |
| 8 | 32 | DLinear | GPT-3.5 | **106.5003** | 107.1156 | 0.6 |
| 8 | 32 | Informer | GPT-3.5 | **106.5003** | 123.6434 | 16.1 |
| 8 | 32 | PatchTST | GPT-3.5 | **106.5003** | 111.4629 | 4.7 |
| 8 | 32 | Reformer | GPT-3.5 | **106.5003** | 118.8497 | 11.6 |
| 8 | 64 | DLinear | GPT-3.5 | **106.3534** | 106.9893 | 0.6 |
| 8 | 64 | Informer | GPT-3.5 | **106.3534** | 124.0805 | 16.7 |
| 8 | 64 | PatchTST | GPT-3.5 | **106.3534** | 115.4039 | 8.5 |
| 8 | 64 | Reformer | GPT-3.5 | **106.3534** | 121.4953 | 14.2 |
| 10 | 32 | DLinear | GPT-3.5 | 108.5963 | **108.4725** | -0.1 |
| 10 | 32 | Informer | GPT-3.5 | **108.5963** | 126.5977 | 16.6 |
| 10 | 32 | PatchTST | GPT-3.5 | **108.5963** | 111.6489 | 2.8 |
| 10 | 32 | Reformer | GPT-3.5 | **108.5963** | 121.0557 | 11.5 |
| 10 | 64 | DLinear | GPT-3.5 | 108.8987 | **108.7955** | -0.1 |
| 10 | 64 | Informer | GPT-3.5 | **108.8987** | 124.6307 | 14.4 |
| 10 | 64 | PatchTST | GPT-3.5 | **108.8987** | 112.8046 | 3.6 |
| 10 | 64 | Reformer | GPT-3.5 | **108.8987** | 119.8280 | 10.0 |
| 12 | 32 | DLinear | GPT-3.5 | **109.1500** | 109.5859 | 0.4 |
| 12 | 32 | Informer | GPT-3.5 | **109.1500** | 126.2477 | 15.7 |
| 12 | 32 | PatchTST | GPT-3.5 | **109.1500** | 111.6083 | 2.3 |
| 12 | 32 | Reformer | GPT-3.5 | **109.1500** | 119.4546 | 9.4 |
| 12 | 64 | DLinear | GPT-3.5 | 109.8162 | **109.6190** | -0.2 |
| 12 | 64 | Informer | GPT-3.5 | **109.8162** | 126.3826 | 15.1 |
| 12 | 64 | PatchTST | GPT-3.5 | **109.8162** | 111.4675 | 1.5 |
| 12 | 64 | Reformer | GPT-3.5 | **109.8162** | 120.6683 | 9.9 |

Table 28: Comparison of Regular (Dynamic Gating) vs. Ablation (Fixed Weights) performance for the Security domain. Increase (%) is relative to the lower MSE in each row.

| Pred Len | Gating Dim | TSF-N Expert | TSF-T Expert | Regular MSE | Ablation MSE | Increase (%) |
|---|---|---|---|---|---|---|
| 6 | 32 | DLinear | GPT-3.5 | **0.8799** | 1.1646 | 32.4 |
| 6 | 32 | Informer | GPT-3.5 | **0.8799** | 1.0627 | 20.8 |
| 6 | 32 | PatchTST | GPT-3.5 | **0.8799** | 1.0628 | 20.8 |
| 6 | 32 | Reformer | GPT-3.5 | **0.8799** | 1.1150 | 26.7 |
| 6 | 64 | DLinear | GPT-3.5 | **0.7651** | 1.1622 | 51.9 |
| 6 | 64 | Informer | GPT-3.5 | **0.7651** | 1.0272 | 34.3 |
| 6 | 64 | PatchTST | GPT-3.5 | **0.7651** | 1.1044 | 44.3 |
| 6 | 64 | Reformer | GPT-3.5 | **0.7651** | 1.1496 | 50.2 |
| 8 | 32 | DLinear | GPT-3.5 | **0.8746** | 1.1247 | 28.6 |
| 8 | 32 | Informer | GPT-3.5 | **0.8746** | 1.0171 | 16.3 |
| 8 | 32 | PatchTST | GPT-3.5 | **0.8746** | 1.0835 | 23.9 |
| 8 | 32 | Reformer | GPT-3.5 | **0.8746** | 1.0778 | 23.2 |
| 8 | 64 | DLinear | GPT-3.5 | **0.7844** | 1.1433 | 45.8 |
| 8 | 64 | Informer | GPT-3.5 | **0.7844** | 1.0262 | 30.8 |
| 8 | 64 | PatchTST | GPT-3.5 | **0.7844** | 1.1089 | 41.4 |
| 8 | 64 | Reformer | GPT-3.5 | **0.7844** | 1.1822 | 50.7 |
| 10 | 32 | DLinear | GPT-3.5 | **0.9079** | 1.2049 | 32.7 |
| 10 | 32 | Informer | GPT-3.5 | **0.9079** | 1.0918 | 20.3 |
| 10 | 32 | PatchTST | GPT-3.5 | **0.9079** | 1.1613 | 27.9 |
| 10 | 32 | Reformer | GPT-3.5 | **0.9079** | 1.2347 | 36.0 |
| 10 | 64 | DLinear | GPT-3.5 | **0.9176** | 1.2457 | 35.8 |
| 10 | 64 | Informer | GPT-3.5 | **0.9176** | 1.1427 | 24.5 |
| 10 | 64 | PatchTST | GPT-3.5 | **0.9176** | 1.1702 | 27.5 |
| 10 | 64 | Reformer | GPT-3.5 | **0.9176** | 1.1758 | 28.1 |
| 12 | 32 | DLinear | GPT-3.5 | **1.0539** | 1.2804 | 21.5 |
| 12 | 32 | Informer | GPT-3.5 | **1.0539** | 1.2125 | 15.0 |
| 12 | 32 | PatchTST | GPT-3.5 | **1.0539** | 1.2227 | 16.0 |
| 12 | 32 | Reformer | GPT-3.5 | **1.0539** | 1.2481 | 18.4 |
| 12 | 64 | DLinear | GPT-3.5 | **0.8885** | 1.2716 | 43.1 |
| 12 | 64 | Informer | GPT-3.5 | **0.8885** | 1.1966 | 34.7 |
| 12 | 64 | Reformer | GPT-3.5 | **0.8885** | 1.2434 | 39.9 |

Table 29: Comparison of Regular (Dynamic Gating) vs. Ablation (Fixed Weights) performance for the SocialGood domain. Increase (%) is relative to the lower MSE in each row.

| Pred Len | Gating Dim | TSF-N Expert | TSF-T Expert | Regular MSE | Ablation MSE | Increase (%) |
|---|---|---|---|---|---|---|
| 6 | 32 | DLinear | GPT-3.5 | **0.1644** | 0.3221 | 95.9 |
| 6 | 32 | Informer | GPT-3.5 | **0.1644** | 0.2908 | 76.8 |
| 6 | 32 | Reformer | GPT-3.5 | **0.1644** | 0.3205 | 94.9 |
| 6 | 64 | DLinear | GPT-3.5 | **0.1632** | 0.3197 | 96.0 |
| 6 | 64 | Informer | GPT-3.5 | **0.1632** | 0.3452 | 111.6 |
| 6 | 64 | Reformer | GPT-3.5 | **0.1632** | 0.3127 | 91.7 |
| 8 | 32 | DLinear | GPT-3.5 | **0.1632** | 0.3040 | 86.3 |
| 8 | 32 | Informer | GPT-3.5 | **0.1632** | 0.2954 | 81.0 |
| 8 | 32 | Reformer | GPT-3.5 | **0.1632** | 0.3330 | 104.0 |
| 8 | 64 | DLinear | GPT-3.5 | **0.1571** | 0.3054 | 94.4 |
| 8 | 64 | Informer | GPT-3.5 | **0.1571** | 0.2955 | 88.1 |
| 8 | 64 | Reformer | GPT-3.5 | **0.1571** | 0.3295 | 109.7 |
| 10 | 32 | DLinear | GPT-3.5 | **0.1597** | 0.3586 | 124.5 |
| 10 | 32 | Informer | GPT-3.5 | **0.1597** | 0.3128 | 95.8 |
| 10 | 32 | Reformer | GPT-3.5 | **0.1597** | 0.3658 | 129.0 |
| 10 | 64 | DLinear | GPT-3.5 | **0.1609** | 0.3462 | 115.2 |
| 10 | 64 | Informer | GPT-3.5 | **0.1609** | 0.3605 | 124.0 |
| 10 | 64 | Reformer | GPT-3.5 | **0.1609** | 0.3432 | 113.3 |
| 12 | 32 | DLinear | GPT-3.5 | **0.2133** | 0.3457 | 62.1 |
| 12 | 32 | Informer | GPT-3.5 | **0.2133** | 0.3120 | 46.3 |
| 12 | 32 | Reformer | GPT-3.5 | **0.2133** | 0.3555 | 66.7 |
| 12 | 64 | DLinear | GPT-3.5 | **0.1978** | 0.3491 | 76.5 |
| 12 | 64 | Informer | GPT-3.5 | **0.1978** | 0.3464 | 75.2 |
| 12 | 64 | Reformer | GPT-3.5 | **0.1978** | 0.3302 | 67.0 |

Table 30: Comparison of Regular (Dynamic Gating) vs. Ablation (Fixed Weights) performance for the Traffic domain. Increase (%) is relative to the lower MSE in each row.

| Pred Len | Gating Dim | TSF-N Expert | TSF-T Expert | Regular MSE | Ablation MSE | Increase (%) |
|----------|-----------|--------------|--------------|-------------|--------------|--------------|
| 6 | 32 | DLinear | GPT-3.5 | **0.0361** | 0.1671 | 363.0 |
| 6 | 32 | Informer | GPT-3.5 | **0.0361** | 0.4357 | 1107.6 |
| 6 | 32 | PatchTST | GPT-3.5 | **0.0361** | 0.2201 | 510.1 |
| 6 | 32 | Reformer | GPT-3.5 | **0.0361** | 0.3776 | 946.6 |
| 6 | 64 | DLinear | GPT-3.5 | **0.0234** | 0.1635 | 599.3 |
| 6 | 64 | Informer | GPT-3.5 | **0.0234** | 0.5500 | 2252.5 |
| 6 | 64 | PatchTST | GPT-3.5 | **0.0234** | 0.2192 | 837.3 |
| 6 | 64 | Reformer | GPT-3.5 | **0.0234** | 0.3794 | 1522.6 |
| 8 | 32 | DLinear | GPT-3.5 | **0.0227** | 0.1256 | 452.4 |
| 8 | 32 | Informer | GPT-3.5 | **0.0227** | 0.4609 | 1926.8 |
| 8 | 32 | PatchTST | GPT-3.5 | **0.0227** | 0.2291 | 907.6 |
| 8 | 32 | Reformer | GPT-3.5 | **0.0227** | 0.3762 | 1554.3 |
| 8 | 64 | DLinear | GPT-3.5 | **0.0194** | 0.1284 | 562.2 |
| 8 | 64 | Informer | GPT-3.5 | **0.0194** | 0.5394 | 2680.8 |
| 8 | 64 | PatchTST | GPT-3.5 | **0.0194** | 0.2291 | 1081.0 |
| 8 | 64 | Reformer | GPT-3.5 | **0.0194** | 0.4447 | 2192.4 |
| 10 | 32 | DLinear | GPT-3.5 | **0.0331** | 0.1807 | 446.5 |
| 10 | 32 | Informer | GPT-3.5 | **0.0331** | 0.5878 | 1677.9 |
| 10 | 32 | PatchTST | GPT-3.5 | **0.0331** | 0.2267 | 585.7 |
| 10 | 32 | Reformer | GPT-3.5 | **0.0331** | 0.3685 | 1014.4 |
| 10 | 64 | DLinear | GPT-3.5 | **0.0281** | 0.1814 | 544.9 |
| 10 | 64 | Informer | GPT-3.5 | **0.0281** | 0.6743 | 2297.8 |
| 10 | 64 | PatchTST | GPT-3.5 | **0.0281** | 0.2223 | 690.4 |
| 10 | 64 | Reformer | GPT-3.5 | **0.0281** | 0.2628 | 834.4 |
| 12 | 32 | DLinear | GPT-3.5 | **0.0215** | 0.1491 | 594.8 |
| 12 | 32 | Informer | GPT-3.5 | **0.0215** | 0.5631 | 2523.3 |
| 12 | 32 | PatchTST | GPT-3.5 | **0.0215** | 0.2240 | 943.7 |
| 12 | 32 | Reformer | GPT-3.5 | **0.0215** | 0.6273 | 2822.5 |
| 12 | 64 | DLinear | GPT-3.5 | **0.0199** | 0.1500 | 654.8 |
| 12 | 64 | Informer | GPT-3.5 | **0.0199** | 0.6402 | 3120.5 |
| 12 | 64 | PatchTST | GPT-3.5 | **0.0199** | 0.2332 | 1073.2 |
| 12 | 64 | Reformer | GPT-3.5 | **0.0199** | 0.2996 | 1407.4 |

Table 31: Comparison of Regular (Dynamic Gating) vs. Ablation (Fixed Weights) performance for the Economy domain. Increase (%) is relative to the lower MSE in each row.

| Expert | Parameters | 1 layer | 2 layers | 4 layers | 6 layers | 8 layers |
|--------|-----------|---------|----------|----------|----------|----------|
| Informer | pred_len=8 | N/A | 0.5063 | **0.4339** | 0.4500 | 0.4761 |
| Informer | pred_len=10 | 0.4097 | 0.4510 | 0.4853 | **0.3493** | 0.4279 |
| PatchTST | pred_len=8 | 0.0325 | 0.0411 | 0.0380 | **0.0220** | 0.0425 |
| PatchTST | pred_len=10 | **0.0232** | 0.0285 | 0.0287 | 0.0376 | 0.0260 |

Table 32: Transformer layer ablation results for the Economy domain (Informer & PatchTST). Bold marks the lowest MSE in each row.

| Expert | Parameters | 1 layer | 2 layers | 4 layers | 6 layers | 8 layers |
|--------|-----------|---------|----------|----------|----------|----------|
| Informer | pred_len=12 | 0.3260 | 0.1907 | **0.1594** | 0.1881 | 0.2574 |
| Informer | pred_len=24 | 0.4247 | 0.3922 | 0.4201 | 0.3983 | **0.2654** |
| PatchTST | pred_len=12 | **0.1128** | 0.1418 | 0.1174 | 0.1206 | 0.1237 |
| PatchTST | pred_len=24 | **0.2410** | 0.2435 | 0.2659 | 0.2686 | 0.2542 |

Table 33: Transformer layer ablation results for the Energy domain (Informer & PatchTST). Bold marks the lowest MSE in each row.

| Expert | Parameters | 1 layer | 2 layers | 4 layers | 6 layers | 8 layers |
|--------|-----------|---------|----------|----------|----------|----------|
| Informer | pred_len=8 | 0.1882 | 0.1571 | **0.1549** | 0.1667 | 0.1569 |
| Informer | pred_len=10 | 0.1643 | 0.1609 | 0.1851 | 0.1693 | **0.1489** |
| PatchTST | pred_len=8 | 0.1782 | 0.1845 | 0.1847 | **0.1718** | 0.1766 |
| PatchTST | pred_len=10 | 0.1870 | 0.1787 | **0.1724** | 0.1824 | 0.1809 |

Table 34: Transformer layer ablation results for the Traffic domain (Informer & PatchTST). Bold marks the lowest MSE in each row.

| Expert | Parameters | 1 layer | 2 layers | 4 layers | 6 layers | 8 layers |
|--------|-----------|---------|----------|----------|----------|----------|
| Informer | pred_len=8 | 0.9227 | 0.7844 | 0.8052 | 1.0007 | **0.7681** |
| Informer | pred_len=10 | 1.0562 | 0.9176 | 0.9045 | **0.8398** | 0.8529 |
| PatchTST | pred_len=8 | **0.9009** | 1.1113 | 0.9324 | 1.0164 | 0.9702 |
| PatchTST | pred_len=10 | 1.0660 | 1.0337 | 1.0652 | **1.0102** | 1.0635 |

Table 35: Transformer layer ablation results for the SocialGood domain (Informer & PatchTST). Bold marks the lowest MSE in each row.

| Expert | Parameters | 1 layer | 2 layers | 4 layers | 6 layers | 8 layers |
|--------|-----------|---------|----------|----------|----------|----------|
| Informer | pred_len=8 | 126.2790 | 124.4728 | **125.0367** | 125.0563 | 124.1039 |
| Informer | pred_len=10 | 128.3774 | 124.4344 | 126.1757 | 125.5975 | **123.1289** |
| PatchTST | pred_len=8 | 108.5710 | 110.1942 | **106.7579** | 107.7070 | 109.8315 |
| PatchTST | pred_len=10 | 110.4228 | 111.2761 | 116.8383 | **109.8659** | 110.2220 |

Table 36: Transformer layer ablation results for the Security domain (Informer & PatchTST). Bold marks the lowest MSE in each row.

| Expert | Parameters | 1 layer | 2 layers | 4 layers | 6 layers | 8 layers |
|--------|-----------|---------|----------|----------|----------|----------|
| Informer | pred_len=12 | 1.0751 | 1.1408 | **0.9890** | 1.0783 | 1.0172 |
| Informer | pred_len=24 | 1.3870 | **1.2082** | 1.3260 | 1.2612 | 1.2829 |
| PatchTST | pred_len=12 | 0.8575 | 0.8395 | 0.8465 | 0.7999 | **0.7931** |
| PatchTST | pred_len=24 | 1.1809 | **1.1338** | 1.1341 | 1.2042 | 1.1350 |

Table 37: Transformer layer ablation results for the Public Health domain (Informer & PatchTST, pred_len = 12, 24). Bold marks the lowest MSE in each row.

| Domain | 1 layer | 2 layers | 4 layers | 6 layers | 8 layers |
|--------|---------|----------|----------|----------|----------|
| Economy | 0.1551 | **0.1779** | 0.2465 | 0.2147 | 0.2431 |
| Energy | **0.2761** | 0.3639 | 0.2407 | 0.2439 | 0.2252 |
| Public Health | 1.2325 | **1.0806** | 1.0742 | 1.0859 | 1.0571 |
| Security | 118.4126 | **114.5918** | 118.7021 | 117.0567 | 116.8216 |
| SocialGood | 0.9865 | 0.9764 | **0.9268** | 0.9668 | 0.9137 |
| Traffic | 0.1794 | 0.1897 | 0.1743 | 0.1725 | **0.1659** |

Table 38: Average MSE across all tested configurations for each number of Transformer layers in the gating module, by domain (Informer & PatchTST only). Bold marks the lowest average per domain.

