# OpenReview forum: "GMM-TS: Gating Architecture for Multi-Modal Time Series Forecasting"
_ICLR.cc/2026/Conference — ICLR 2026 Conference Withdrawn Submission_

### Official Review · Reviewer_KGZs · 2025-10-25

**Soundness:** 2
**Presentation:** 2
**Contribution:** 2
**Rating:** 2
**Confidence:** 4

**Summary:**

This paper introduces GMM-TS, a Transformer-based gating architecture for multi-modal time series forecasting. GMM-TS dynamically integrates predictions from multiple uni-modal experts (e.g., numerical and textual), leveraging a learnable, per-time-step gating mechanism using a Transformer encoder. The framework supports more than two experts, offers both joint and offline training, and is claimed to provide modularity and interpretability. The approach is evaluated on the TimeMMD benchmark across nine domains and four forecast horizons, with comprehensive ablations and interpretability demonstrations suggesting improved performance over contemporary baselines.

**Strengths:**

1. GMM-TS elegantly generalizes the mixture-of-experts framework to handle an arbitrary number of heterogeneous experts and modalities, extending beyond the binary, static fusion in existing multi-modal TSF methods. The approach notably supports numerical and textual modalities, but its design enables future extensions.

2. The use of a Transformer-based gating network facilitates fine-grained, input-adaptive expert weighting at every time step. The gating weights provide inherent interpretability, enabling users to visualize per-expert contributions.

**Weaknesses:**

1. Although the paper claims that integrating heterogeneous data types remains underexplored, several recent studies have already proposed advanced multimodal time series forecasting frameworks [1–4]. In comparison, the proposed approach constitutes a simple fusion strategy without introducing fundamentally new mechanisms or insights into multimodal alignment or cross-domain interaction.

2. Related works such as T3Time [5] and Multi-Resolution Time-Series Transformer [6] also employ dynamic gating, residual alignment, and multi-resolution fusion, yet are omitted from the Related Work section.

3. Despite highlighting support for more than two experts and the extensibility to new modalities, all quantitative experiments involve only numerical and textual data. The claimed scalability to other modalities (e.g., vision) is not empirically validated.

4. Interpretability is primarily inferred from expert-weight dynamics (Fig. 3) and restricted to parameter visualizations. However, such visual inspection provides little actionable insight or transferable understanding for the community. Without quantitative or causally grounded analyses, the presented interpretability remains descriptive rather than explanatory.

5. The gating Transformer has substantial capacity relative to dataset size and forecast horizons, yet training/validation learning curves and data-scarcity analyses are absent. Robustness to small datasets or poorly calibrated experts is not examined.

6. Several key design choices—such as expert architecture selection, pooling and regularization strategies, and handling of forecast-horizon mismatches—are insufficiently detailed to ensure replicability.

7. No quantitative evaluation against the related approaches discussed in Weaknesses 1 and 2 is provided, leaving it unclear whether the proposed framework—beyond its gating design—offers tangible advantages over existing multimodal, multi-branch, or multi-resolution forecasting architectures.


[1] Time-VLM: Exploring Multimodal Vision-Language Models for Augmented Time Series Forecasting, ICML 2025

[2] Timecma: Towards llm-empowered multivariate time series forecasting via cross-modality alignment, AAAI 2025

[3] Autotimes: Autoregressive time series forecasters via large language models, NeurIPS 2024

[4] Time-llm: Time series forecasting by reprogramming large language models, ICLR 2024

[5] T3time: Tri-modal time series forecasting via adaptive multi-head alignment and residual fusion, arXiv 2025

[6] Multi-resolution time-series transformer for long-term forecasting, AISTATS 2024

**Questions:**

See Weaknesses.

---

### Official Review · Reviewer_kvaX · 2025-10-27

**Soundness:** 2
**Presentation:** 2
**Contribution:** 1
**Rating:** 2
**Confidence:** 3

**Summary:**

This paper introduces GMM-TS, a learnable gating architecture for multi-modal time series forecasting inspired by Mixture-of-Experts.
The model employs a Transformer Encoder to dynamically compute per-time-step weights, allowing it to adaptively fuse predictions from any number of specialized uni-modal experts (e.g., text and numerical).
This modular and interpretable framework was extensively evaluated on the TimeMMD benchmark. The results demonstrate that GMM-TS consistently outperforms state-of-the-art baselines, including TimeMMD and GPT4MTS, across nine domains and multiple forecast horizons.

**Strengths:**

1. The paper's srength is its novel application of a gating mechanism, inspired by MoE, to the complex problem of multi-modal fusion . Instead of just selecting one expert, the architecture learns to blend heterogeneous experts (e.g., numerical TSF-N and textual TSF-T). To the best of my knowledge, this is the first framework to effectively fuse more than two experts (e.g., two numerical, one text), demonstrating significant originality and scalability beyond the binary fusion limitations of prior work .

2. A major advantage of this gating approach is its dynamic, per-time-step weighting. Unlike prior work using static or fixed weights (like TimeMMD), GMM-TS can adaptively prioritize the most relevant expert at each specific step of the forecast horizon. This is strongly validated by the ablation study (Table 26), which shows that removing this dynamic capability leads to a catastrophic performance drop .

3. The GMM-TS framework directly tackles the key limitations of existing multi-modal TSF methods . By producing explicit, per-expert weights, it partially (weaknesses) solves the interpretability problem of implicit LLM-based fusion (like GPT4MTS) , as demonstrated in Figure 3 . Furthermore, its "plug-and-play" design provides the modularity and flexibility (supporting both joint and offline training) that monolithic architectures lack .

**Weaknesses:**

1. **Limited Novelty over TimeMMD**: The core contribution, while effective, appears to be an incremental extension of the TimeMMD framework. The paper adopts TimeMMD's overall architecture and its specific formulation for handling TSF-T experts (using a frozen LLM and a trainable MLP) . The primary novelty is the replacement of TimeMMD's static fusion weight with a dynamic gating network. While this is a meaningful improvement, it makes the work feel more like a refinement of a prior method rather than a fundamentally new architecture.

2. **Insufficient Scope of Experimental Validation**: The experimental validation, while thorough against TimeMMD , feels somewhat narrow in scope. The comparative analysis is heavily focused on TimeMMD, and the paper would be significantly stronger if it benchmarked GMM-TS against a wider range of modern, high-performing uni-modal and multi-modal baselines (e.g., TimeXer). This concern about expert selection also applies to the internal components. The analysis would be more compelling if it focused exclusively on fusing current state-of-the-art experts to demonstrate the best-case potential of the gating mechanism.

3. **Insufficient Detail on Text Modality Processing**: A significant weakness is the paper's lack of self-contained detail on how the textual data ($X_t$) is processed. The paper states that it "follows the TimeMMD protocol" for prompting and handling the TSF-T experts. While referencing prior work is standard, the text-numerical fusion is central to this paper's algorithm. By delegating this core methodological explanation to another paper, it forces the reader to seek external sources to understand a critical component of the proposed model, hindering the paper's clarity and completeness.

4. **Superficial Analysis of the Gating Mechanism**: The paper's main novelty is the dynamic, multi-expert gating. However, the analysis of this core mechanism is limited to two brief visualizations in Figure 3 . This is insufficient to fully understand how the model learns to assign weights. For example,
- How gating weights vary across different domains (e.g., does the model trust text more in 'Economy' than in 'Climate'?).
- How weights evolve over longer forecast horizons (by statistical analysis).
- What specific input patterns (e.g., a spike in the numerical data or a keyword in the text) trigger a shift in the gating weights.

Without this analysis, the gating network remains a bit of a "black box," which undermines the paper's strong claims of interpretability.

**Questions:**

Please refer to weaknesses

---

### Official Review · Reviewer_cea4 · 2025-10-28

**Soundness:** 3
**Presentation:** 2
**Contribution:** 2
**Rating:** 4
**Confidence:** 4

**Summary:**

This paper proposes **GMM-TS**, a Transformer-based **gating architecture** for multi-modal time-series forecasting.
The model dynamically fuses predictions from multiple unimodal experts (e.g., numerical, textual) using a learnable gating mechanism inspired by **Mixture-of-Experts (MoE)**. Unlike prior work such as **TimeMMD** and **GPT4MTS**, which rely on static or implicit fusion, GMM-TS performs **fine-grained, interpretable, input-dependent fusion** and supports more than two modalities.  The paper reports extensive experiments on the **TimeMMD benchmark** covering nine domains and multiple forecasting horizons. Results show consistent improvements over strong baselines and demonstrate interpretable per-time-step expert weights. Ablation studies confirm robustness to architectural choices and highlight the necessity of adaptive gating.

**Strengths:**

1. **Methodological clarity and motivation** — The paper clearly identifies the limitations of existing multi-modal forecasting systems and introduces a gating architecture that is conceptually simple yet practically effective.
2. **Comprehensive experiments** — The evaluation spans nine domains and 540 runs, including both uni-modal and multi-modal baselines, which makes the evidence convincing.
3. **Interpretability** — Visualization of expert weights over time effectively shows how the model prioritizes different modalities under changing conditions.
4. **Modularity and extensibility** — The framework can integrate arbitrary expert combinations and supports both joint and offline training.
5. **Readable writing and solid empirical grounding** — The structure is clear, and the methodology is well justified.

**Weaknesses:**

1. **Limited novelty beyond MoE adaptation** — The main contribution lies in applying and extending mixture-of-experts ideas to multi-modal forecasting; conceptually, this is more incremental than revolutionary.
2. **Restricted modality diversity** — Experiments are limited to textual and numerical data; no visual or sensor modalities are tested.
3. **Scalability discussion is shallow** — The Transformer-based gating mechanism could become computationally expensive when the number of experts grows; this is not empirically explored.
4. **Comparisons with recent baselines (e.g., MERA, Time-LLM, Time-VLM)** are deferred to the appendix without quantitative results in the main paper.
5. **Ablation clarity** — Some ablation tables are long and repetitive; the core takeaways could be summarized more directly.

**Questions:**

1. How does the computational complexity of the gating mechanism scale with the number of experts?
2. Can GMM-TS handle missing modalities (e.g., when text data is unavailable during inference)?
3. How stable is the joint training strategy compared to offline pretraining when using larger LLM experts?
4. Are the gating weights conditioned only on latent features, or also on training-time prediction errors?
5. How might the model behave under non-stationary temporal regimes or domain shifts?

---

### Official Review · Reviewer_JmrS · 2025-10-31

**Soundness:** 3
**Presentation:** 2
**Contribution:** 3
**Rating:** 4
**Confidence:** 3

**Summary:**

This paper proposes a Transformer-based gating architecture called GMM-TS to address multi-modal time series forecasting problems. Inspired by the mixture-of-experts architecture, GMM-TS uses a trainable gating network to determine the weights of various experts, dynamically combining them to solve multi-modal time series forecasting problems for different tasks.

**Strengths:**

•	This paper Proposes the first modular and interpretable gating mechanism for multi-modal time series forecasting, extending Mixture-of-Experts principles to heterogeneous modalities.

•	This paper includes ablation studies on aggregation strategies, gating dimensions, and adaptive vs. static gating, as well as comparisons of joint vs. offline pretraining. The experimental section of the paper is very comprehensive.

•	The proposed GMM-TS architecture supports the dynamic fusion of multiple experts and provides some interpretability to the prediction results through weight distribution.

**Weaknesses:**

•	The core innovation of the paper seems unclear, and the core contribution of GMM-TS lacks novelty. It seems more like a simple concatenation of multimodal time series prediction with MoE.

•	The combination mechanism of the TSF-N and TSF-T frameworks is not explained in further detail, and the training part of the gating network lacks more description.

•	Although GMM-TS aims to solve the time series prediction problem in the multi-modal domain, the actual framework and experiments only cover the common text and time series domains. Its true usability and portability for multi-modal applications still need further experimental verification.

•	The lack of further ablation experiments on the gating network and the input sequence to verify the rationale for the gating network leaves the actual effect of the gating network and its verifiability.

•	The actual inference efficiency of gating networks and the actual inference latency they introduce have not been experimentally compared.

**Questions:**

•	Could you provide more modal input data, as well as comparative data on inference efficiency and inference accuracy across more scenarios?

•	When more experts are incorporated into the framework, how can we avoid expert redundancy and overfitting, and how can we ensure that the gating network can accurately provide better choices?

•	When a single expert performs poorly on a given task, can multiple experts within the GMM-TS framework achieve a degree of complementarity to achieve better performance?

---

### Note · Authors · 2025-11-27

I have read and agree with the venue's withdrawal policy on behalf of myself and my co-authors.